# SokoBench: Evaluating Long-Horizon Planning and Reasoning in Large Language Models

Sebastiano Monti                    *s.monti@ipazia.com Ipazia SpA, Italy*

Carlo Nicolini                    *c.nicolini@ipazia.com Ipazia SpA, Italy*

Giovanni Pellegrini                    *g.pellegrini@ipazia.com Ipazia SpA, Italy*

Jacopo Staiano                    *jacopo.staiano@unitn.it University of Trento, Italy*

Bruno Lepri                    *lepri@fbk.eu Fondazione Bruno Kessler and Ipazia SpA, Italy*

**Reviewed on OpenReview:** *https://openreview.net/forum?id=pLosAkOoGU*

## Abstract

Although the capabilities of Large Language Models and Large Reasoning Models have been increasingly tested on complex reasoning tasks, their long-horizon planning abilities have not yet been extensively investigated. In this work, we provide a systematic assessment of the planning and long-horizon reasoning capabilities of state-of-the-art Large Reasoning Models (LRMs). We propose a novel benchmark based on Sokoban puzzles, intentionally simplified to isolate long-horizon planning from state persistence. Our findings reveal a consistent degradation in planning performance when more than 25 moves are required to reach the solution, suggesting non-recoverable error accumulation under single-pass autoregressive decoding. We show that equipping LRMs with Planning Domain Definition Language (PDDL) parsing, validation, and solving tools allows for modest improvements, suggesting that character level counting and long yet simple state tracking might not be overcome by test-time scaling approaches alone.

## 1 Introduction

Automated Planning, i.e. the task of generating sequences of actions to achieve a goal, is a well-studied problem in the field of Artificial Intelligence (AI) (Ghallab et al., 2016), since it requires AI systems to exhibit cognitive abilities such as reasoning, understanding, and efficient state space search (Wei et al., 2025). To this end, automated planning literature has focused on the use of formal languages, such as the Planning Domain Definition Language (PDDL) (McDermott et al., 1998; Russell & Norvig, 2021; Haslum et al., 2019)), and of tree-search strategies or specific heuristics to find optimal solutions (Bonet & Geffner, 2001). Large Language Models (LLMs) and, in particular, Large Reasoning Models (LRMs) i.e., LLMs trained to produce so-called *reasoning traces* resembling structured thought processes, have demonstrated impressive capabilities in natural language understanding, knowledge retrieval and multi-modal pattern recognition (Jaech et al., 2024; Guo et al., 2025; Team et al., 2025). However, recent studies highlighted the limitations of such models when applied to planning tasks (Valmeekam et al., 2023b; Shojaee et al., 2025). For instance, internal reasoning processes have been shown to resemble a form of wandering through the solution space rather than a systematic exploration (Lu et al., 2025). This distinction becomes particularly important for problems that require maintaining sequential state information, such as spatial exploration in constrained environments. In these settings, effective tracking of working memory is necessary to infer the agent's latent previous state (Zhang et al., 2024).

In this work, we investigate the long-horizon planning abilities of LRMs using a highly simplified variant of the Sokoban puzzle (Culberson, 1998). Rather than increasing spatial complexity, we *deliberately minimize*

*the structural complexity* of the environment while preserving the long-horizon nature of the task by creating examples with the lowest possible branching factor compatible with solvability: a single movable block, one goal and one player placed within a linear corridor with tightly controlled geometry. In addition to evaluating reasoning-only configurations, we introduce an LLM-modulo variant in which the model must generate a full PDDL encoding of the corridor instance that is subsequently solved by a dedicated classical planner. This hybrid configuration allows us to decouple plan generation from plan execution and to quantify how much of the failure originates from internal reasoning versus formal problem specification. Furthermore, we systematically evaluate the effect of explicit chain-of-thought prompting across reasoning and non-reasoning models (see Appendix E). Finally, to assess whether the observed phenomena generalize beyond Sokoban, we replicate the same controlled scaling philosophy on a simplified Klotski-style environment, which we call *Freetski* (see Appendix G).

These settings allow us to isolate long-horizon planning from state persistence: models are required to produce complete solution sequences without external memory, intermediate feedback, or state validation, relying solely on internal state representations to track the evolving environment. In the reasoning-only regime, solutions must be generated in a single forward pass; in the LLM-modulo regime, correctness depends on the accuracy of the symbolic encoding provided to the solver. We therefore investigate to what extent LRMs can sustain coherent planning over long (but simple) action sequences and whether even minimal reasoning branching in otherwise trivial Sokoban instances is sufficient to induce planning failures.

Concretely, we examine whether current LRMs can reliably solve linear-corridor Sokoban puzzles with minimal possible branching and identify the point at which increases in horizon length lead to catastrophic breakdowns in action validity, despite the simplicity of the underlying environment. These minimal sub-problems, which are trivial to humans (Jarušek & Pelánek, 2010), remain challenging for Large Reasoning Models, consistent with previous findings on spatial intelligence limitations (Cai et al., 2025). With a systematic assessment of multiple ablations studies, from solver-assisted LRM-modulo experiments, to controlled Klotski-style extensions, chain-of-thought prompting, and a full breakdown on maps rotations, we provide an analysis of long-horizon stability under tightly constrained geometric conditions.

## 2 Related Work

### 2.1 Benchmarks for LLM Planning

As mentioned above, planning requires LLMs to blend logical, numerical, and spatial reasoning with long-horizon strategic adaptation, rather than just relying on pattern matching or memorization. Classical planning domains expressed in or derived from the Planning Domain Definition Language (PDDL (Fox & Long, 2003)), such as BlocksWorld (Slaney & Thiébaux, 2001), Towers of Hanoi and similar tasks (Pallagani et al., 2023), remain a common benchmark choice, though earlier attempts date back to the pre-ChatGPT era (Silver et al., 2022). Test suites like PlanBench (Valmeekam et al., 2022) introduced structured, domain-agnostic evaluations inspired by classical planning (Ghallab et al., 2016), including plan generation (Oswald et al., 2024; Valmeekam et al., 2025; La Malfa et al., 2025) and optimality (Valmeekam et al., 2022; Zhai et al., 2025; Valmeekam et al., 2023a).

In another line of work, planning is evaluated within *agentic* or workflow-based frameworks, where LLMs are required to decompose goals into multiple sub-plans (Meyerson et al., 2025; Zhang et al., 2025; La Malfa et al., 2025). The results in these settings are encouraging though highly cost intensive. Importantly, when not equipped with external tools or made part of larger workflows (e.g., enabling stateful tracking (Hu et al., 2025b)), innate planning abilities remain still weak (Schepanowski & Ling, 2025). Even the latest foundational models are found to consistently fail in delivering correct sequences of actions (in any format or language) due to two primary deficits: weak internal state representations leading to invalid moves and misleading heuristic search resulting in loops or early termination, as shown in the textual game "8-puzzle" in Schepanowski & Ling (2025). Moreover, efficacy of different prompting techniques is model-dependent in a non-predictable way (Schepanowski & Ling, 2025; Deng et al., 2025).

Other works have systematically investigated the performances of LLMs in playing textual games with *gym*-style APIs (Brockman et al., 2016; Hu et al., 2025a). Beyond structured puzzles, community-driven and

informal game-oriented benchmarks like word-game bench (Stojanovski, 2024) and nonogram logic puzzles (Berend et al., 2014; Kleine, 2026) with multi-difficulty instances have been devised to measure how well models plan under both explicit and implicit constraints, track environment states, and adapt over multiple turns. The varying depth of planning ability required helps to reveal how performance scales with complexity and structure.

In general, existing benchmarks using specific planning languages and/or internal reasoning traces expressed in natural language show that LLMs exhibit limited planning abilities in various domains (Kambhampati et al., 2024), especially as the complexity and horizon length of the problems increase. This gap motivates the development of new benchmarks tailored to planning and solving structured textual puzzles with LLMs.

## 2.2 Sokoban as a Benchmark for Planning

The Sokoban puzzle involves spatial planning in a highly constrained environment. Solvable Sokoban maps can be generated efficiently (Murase et al., 1996), and the environment is fully controllable and deterministic. These properties enable rigorous evaluation using exact solvers and verifiers, as well as metrics such as search depth and solution time (Jarušek & Pelánek, 2010; Shoham & Schaeffer, 2020). Unlike puzzles such as the Tower of Hanoi, which can be solved by repeating a simple pattern for larger instances, Sokoban offers no shortcuts. Each map is unique, and moving a single box can block or open paths in ways that prevent a one-size-fits-all solution. As a result, Sokoban is considered a good benchmark for evaluating planning abilities in the 2023 edition of the International Planning Competition (Taitler et al., 2024). Sokoban has also been widely adopted in recent works as a benchmark for evaluating reasoning and planning in LLMs, often using heterogeneous map distributions or fixed difficulty tiers (Hu et al., 2025a; Wang et al., 2025). On realistic maps most recent models like `claude-3.7` or `openai-o3` never exceed 10% solved instances, with most of them totally failing.

Recently, recurrent neural networks (not LLM-based) trained over multiple examples of Sokoban puzzles have obtained state of the art performance (Jolicoeur-Martineau, 2025; Taufeeque et al., 2024). However, LLMs are found to perform poorly, struggling even with simple maps and correctly solving only a small fraction of instances: Valmeekam et al. (2025) report success rates of just about 10–12% when using the OpenAI `o1-preview` model directly. In contrast, substantially higher success rates are achieved in an LLM-Modulo setting, where the same model is used to generate plans that are then executed by an external planner, yielding approximately 43% solved instances for `o1-preview` (and about 10% for `o1-mini`), albeit at significantly higher computational cost. Although these results show that Sokoban is a demanding benchmark for LLM-based planning, most prior work reports overall success rates without systematically varying geometric symmetries or extending the planning horizon beyond training-like instances. We instead examine how performance declines as a function of one single dimension and how it changes under map rotations, focusing on invariance and extrapolation rather than aggregate solve rate alone.

Most prior work on textual puzzle solving and planning with LLMs has emphasized high-level notions such as search depth, branching factor, or overall puzzle complexity. Much less attention has been paid to the role of simpler, low-level operations that these tasks implicitly rely on. Evidence from seemingly trivial problems suggests that LLM failures do not always stem from complexity itself, but from how basic reasoning steps are elicited. A well-known example is the character-counting question "how many r's are in strawberry?" (Karpathy, 2024), which has sparked debate over whether LLM errors are caused by tokenization or deeper representational limits (Shin & Kaneko, 2024). The work by Xu & Ma (2025) revisits this issue through a careful empirical study, showing that LLMs are in fact capable of performing these simple symbolic operations, but often fail unless prompted to reason explicitly. Character-level benchmarks, such as CharBench (Uzan & Pinter, 2025), show that modern LLMs struggle with simple character counting and positioning tasks not because tokenization fully explains these errors, but because intrinsic properties like word length and actual character count have a stronger influence on performance, indicating that basic symbolic operations are not reliably deployed unless the model is guided to engage them explicitly.

Taken together, these findings motivate a more fine-grained analysis of planning failures. Rather than attributing errors solely to global puzzle complexity, we examine whether breakdowns emerge from horizon length, geometric transformation, or problems in low-level operations such as counting, state tracking, and

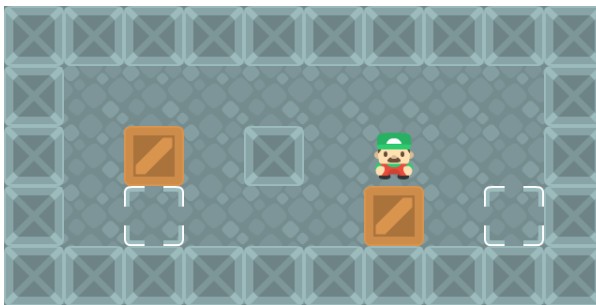

Figure 1: Example of a Sokoban puzzle. All crates must be pushed onto goal positions (white borders). A solution to this problem in compressed notation is: 1↑, 4←, 1↓, 1→, 1↓, 4→, resulting in the LURD notation `u,l,l,l,l,D,r,d,r,r,R,R`.

**Equivalent ASCII format**

```
# # # # # # # # # #
#                 #
#   $   #   @     #
#   .       $   . #
# # # # # # # # # #
```

| Game | Map Element | ASCII Symbol |
|---|---|---|
| | Player | @ |
| | Player on Goal | + |
| Sokoban | Box | $ |
| | Box on Goal | * |
| | Goal | . |
| | Wall Brick | # |

Table 1: ASCII notation of the elements of Sokoban maps. Empty areas are encoded as space (␣).

directional consistency. By situating our corridor-based Sokoban study within this broader line of work, we aim to clarify which aspects of LRM planning behavior are domain-specific and which reflect more general limitations in long-horizon structured reasoning.

## 3 Methods

### 3.1 Sokoban game

Figure 1 shows an example of a Sokoban puzzle and the game's central mechanic: the player controls a sprite that pushes boxes within a two-dimensional spatially constrained environment with the goal to position them onto predefined locations. Despite its apparent simplicity, Sokoban is a NP-hard and PSPACE-complete problem (Culberson, 1998), positioning it as a canonical domain for symbolic and hierarchical planning.

Apart from the pictorial representation, Sokoban maps can be encoded using an ASCII-based symbolic representation as expressed in Table 1. Sequences of main character actions are typically encoded in LURD format (left, up, right, down), with lowercase letters indicating simple moves, and uppercase letters indicating box pushes. Although moves and pushes have distinct notations in classical Sokoban planning, in our experiments we restrict only to comma-separated uppercase letters. This representation does not compromise the information content of the solutions and simplifies the output format for language models, avoiding potential mistakes.

### 3.2 Dataset

We generated a dataset consisting of narrow, corridor-like maps, i.e. maps of width $\ell$ and height 1. Each map contains the same set of elements: one player, one box, and one goal. The maps share the same initial configuration in which the goal is positioned at one end of the map, the player at the opposite end, and the box placed in between the two, so that all elements lie along the same row or column. This choice is motivated by its simplicity: the corridor length $\ell$ is the only map parameter and it serves as a proxy for map difficulty. Hence, with just one degree of freedom to account for, we overcome the problem of defining complex measures for solution difficulty: the longer the map, the harder the task.

In our benchmark, we consider map lengths $\ell$ ranging from 5 to 100 in increments of 5. For each map, we generate four augmented variants corresponding to rotations of 90°, 180°, and 270°, as well as the original (unrotated) orientation. This augmentation strategy reduces the risk of querying the model with data that may have been encountered during pretraining and enables analysis of whether models exhibit orientation-dependent performance. In total, the evaluation set comprises 80 distinct maps, spanning 20 values of $\ell$ with

four orientations each. We publicly release our dataset at `https://huggingface.co/datasets/Linello/sokobanlevels`.

### 3.3 Experimental Setup

We employ both open and closed weights models, specifically `deepseek-r1-0528` (Guo et al., 2025), `gpt-5-mini` (OpenAI, 2025b), `gpt-oss-120b` (OpenAI, 2025c), `grok-4.1-fast` (xAI Team, 2025) and `gemini-3-pro` (Google DeepMind, 2025). They are all *reasoning models*, i.e., they are configured to generate an explicit reasoning trace prior to emitting the final answer to the user query. For GPT models, we don't change the default temperature neither the default reasoning effort (set to medium). Instead we cap the maximum number of completion tokens (including both reasoning and final answer tokens) at 32,768. All inference calls are routed through OpenRouter,[1] with the inference provider consistently set to DeepInfra.[2] Apart from maximum token limit, we leave the default OpenRouter settings for all the models.

#### 3.3.1 One-shot Inference

In the first experimental setup, we test the ability of the selected LRMs to solve simple Sokoban puzzles when provided only with the instructions, the mapping of characters as in Table 1 and a single demonstration. Under this setup, models are by design limited to use exclusively their internal state representations to solve Sokoban puzzles of varying solution lengths. The prompts used for all models are described in Appendix A.

#### 3.3.2 LRM-Modulo

In the second experimental setup, we investigated if and how Sokoban puzzle–solving performance can be enhanced when LRMs are provided with access to external planning solvers, within an LRM-modulo framework analogous to that of Valmeekam et al. (2023a). To this end, we prompted the models to generate specific instances of planning problems while providing them with a pre-existing, human-authored and verified PDDL domain (Appendix B.2). In this setup, the model is responsible solely for formulating the PDDL problem, which is then processed through an agentic pipeline. This workflow utilizes a domain parser to instantiate the formal world representation and a dedicated problem parser that acts as a validator, informing the model whether the generated problem is syntactically and semantically well-formed. Finally, the pipeline provides access to specialized PDDL planners such as `Fast-Downward` or `PyperPlan`, integrated via the `Unified Planning` library (Micheli et al., 2025; Alkhazraji et al., 2020; Helmert, 2006) to solve the problem and get the optimal plan. All the tools were wrapped and made accessible to the LRMs via a custom Model Context Protocol library (Anthropic, 2024) implemented with `FastMCP` library (Lowin, 2024). The design of our architecture is shown in Figure 2.

The planner tool produces a variety of diagnostic and informational messages that are provided back to the model, including error reports, timing information, and the complete raw response. This raw response can be further processed to extract the LURD solution in cases where the problem is successfully solved. In failure scenarios, the tool returns the encountered errors in natural language to the LRM. Errors or warnings are generated in situations such as logically inconsistent or unsatisfiable problems, invalid or inappropriate initial conditions, or when the solver exceeds the maximum allotted execution time (60 seconds). The agentic loop ends either with a valid plan or with a message to the final user explaining that, after three failed attempts (which may include having the LRM reformulate the PDDL problem), the agent could not find a satisfactory solution.

The LRM-modulo pipeline is considerably slower than the reasoning-only one. It took an average of 75 minutes using `gpt-5-mini` on an AWS `t3.xlarge` instance (4 CPUs at 3.1 GHz, 16 GB RAM) to collect the points shown in Figure 6a and more than four hours in the case of `gemini-3-pro`. The prompts being used for the LRM-Modulo experiments are described in Appendix B. A full breakdown of the performance at different map rotations is illustrated in Appendix C.

---

[1] https://openrouter.ai/
[2] https://deepinfra.com/

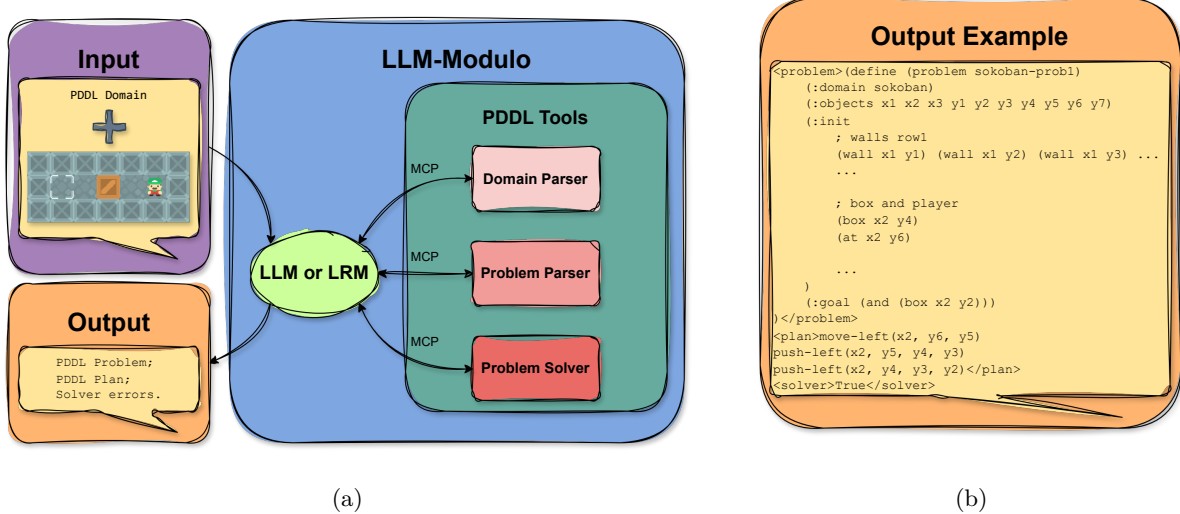

Figure 2: Panel (a) represents a simple schema of our LRM-modulo pipeline. The detailed input prompts are collected in Appendix B.1, while an example of output is shown in Panel (b).

## 3.4 Evaluation

A solution to a Sokoban instance is defined as a sequence of actions that transforms the system from its initial configuration to the final state, where all boxes are correctly placed on goal positions. Multiple valid solutions may exist for the same map, however we restrict the evaluation only to optimal solutions, i.e., sequences that achieve the goal with the minimum possible number of moves. The intrinsic simplicity of our setting makes optimality the natural criterion. Clearly, the one-dimensional layout of the map allows only for a unique optimal solution: allowing arbitrary exploratory trajectories to count as success would effectively reward environment-style trial-and-error behavior rather than structured forward reasoning, which is not the purpose of our analysis.

**Accuracy:** Given a map of length $\ell$, we define the accuracy in Eq. 1 as the expectation, over all repetitions and rotations $N$ of the indicator of exact string equality (via Iverson brackets) between the predicted action sequence $\hat{\mathbf{x}}^{(\ell)}$ and the ground-truth sequence $\mathbf{x}^{(\ell)}$, i.e., the fraction of trials in which the two character strings are identical:

$$\text{Accuracy}(\ell) = \frac{1}{N} \sum_{n=1}^{N} [\hat{\mathbf{x}}^{(\ell)} = \mathbf{x}^{(\ell)}]. \tag{1}$$

Here $N$ is the product of the total number of trials $n_t$ and the number of map rotations $n_r = 4$. Increasing $n_t$ mitigates the intrinsic non-determinism of the obtained solutions, by sampling at multiple seeds. In the LRM one-shot experiments, we set the number of repetitions $n_t$ to 8. Conversely, in the LRM-modulo experiments, the substantially higher computational and monetary costs imposed stricter constraints. We therefore reduced the number of repetitions $n_t$ to 4.

**Prefix accuracy:** Alongside the standard accuracy metric, we define Prefix Accuracy (Eq. 2) to provide a more granular evaluation of model performance. This metric calculates the average proportion of correct symbols generated by comparing the predicted and true plans' strings element-wise:

$$\text{PrefixAccuracy}(\ell) = \frac{1}{N} \sum_{n=1}^{N} \frac{[m^{(n)} \leq \ell]}{\ell} \sum_{i=1}^{m^{(n)}} [\hat{x}_i^{(n)} = x_i^{(n)}], \tag{2}$$

where $m^{(n)}$ is the length of the predicted plan $\hat{\mathbf{x}}^{(n)}$. Unlike the hard matching of the standard accuracy metric, prefix-accuracy is more optimistic, rewarding the model for correct partial trajectories even if it stops prematurely. However, it remains strictly penalized for overshooting: if the predicted length $m^{(n)}$ exceeds the ground-truth length $\ell$, the score for that trial is 0. For instance, a prediction $\hat{\mathbf{x}}^{(n)} = (\mathtt{l}, \mathtt{l}, \mathtt{l})$ against a ground truth $\mathbf{x}^{(n)} = (\mathtt{l}, \mathtt{l}, \mathtt{l}, \mathtt{l})$ yields a score of $3/4$, whereas any prediction exceeding length 4 results in a score of 0.

**Manhattan Distance:** While string-based metrics evaluate the symbolic fidelity of the action sequence, they do not account for the spatial proximity of the agent to the objective. We therefore use the Manhattan Distance (Eq. 3) to measure the $L_1$ distance between the agent's terminal position and the goal, independent of sequence semantics or environmental obstacles. This metric, different from the accuracy with its $0/1$ binary output, can provide a better idea of the adherence of the final cumulated plan result as the distance from the target, highlighting potential problems like physical constraints break or wandering behaviors.

$$D(\ell) = \frac{1}{N} \sum_{n=1}^{N} \left( |x_{\text{final}}^{(n)} - x_{\text{goal}}^{(n)}| + |y_{\text{final}}^{(n)} - y_{\text{goal}}^{(n)}| \right) \tag{3}$$

Here, $(x_{\text{final}}^{(n)}, y_{\text{final}}^{(n)})$ represents the agent's coordinates after executing all moves in the predicted sequence $\hat{\mathbf{x}}^{(n)}$, starting from the origin $(0, 0)$. The goal coordinates $(x_{\text{goal}}^{(n)}, y_{\text{goal}}^{(n)})$ are always at a fixed distance $\ell$ from the origin, specifically $(\pm\ell, 0)$ for $0°/180°$ rotations and $(0, \pm\ell)$ for $90°/270°$ rotations.

The primary motivation for this metric is to distinguish between "near-misses" and total navigational failures. By measuring spatial displacement, we can quantify whether a model that failed the exact string match nonetheless moved in the correct direction or reached the vicinity of the goal. This provides a soft failure signal that string-based metrics like Accuracy or Prefix Accuracy cannot capture.

## 4 Results

### 4.1 One-shot Inference

Figure 3 summarizes the results in terms of accuracy and total token usage. The plot on the left of Figure 3 shows the accuracy as a function of the corridor length, $\ell$, for all tested models. Similarly to Shojaee et al. (2025), our results show approximately three regions in which the models behave according to different regimes: an easier region where corridor lengths are short, characterized by higher accuracy; an intermediate region characterized by a rapid decrease in accuracy as the length of the corridors increases and a harder region in which the models completely fail to return a correct plan. These regions are specific to each model.

Crucially, corridors are *deep but narrow* problems: many sequential steps (depth $d \sim \ell$) with minimal branching. In such settings, a small per-step probability $p_w$ of miscounting the size of the map compounds exponentially, yielding success probability $\sim (1 - p_w)^\ell$. This may explain the three-region performance curve we observe: short corridors tolerate occasional drift, producing a plateau of acceptable accuracy; intermediate lengths mark the onset of exponential decay, while long corridors see near-total collapse as cumulative errors dominate. We thus believe that the main reason LRMs cannot correctly plan in longer corridors is mainly due to internal counting representation. It was indeed shown in McCoy et al. (2024) that when asked to count individual characters, LLMs perform better with common characters than uncommon ones (like `#`). This counting failure can be interpreted through the lens of Lu et al. (2025) "wandering vs systematic exploration" framework: maintaining an accurate count over many positions is equivalent to maintaining correct state representations across a chain of transitions.

As an observation, we report that `gpt-5-mini` displays an anomalous accuracy peak around $\ell = 50$ which is however hardly explained by the model above. We believe this effect is likely due to memorization, but without access to internal states models this remains an hypothesis.

Figure 3 shows the number of output tokens as a function of the corridor length, $\ell$, filtering only for correctly solved Sokoban problems. Linear regression analysis reveals that for each model, the number of output tokens

for correctly predicted problems increases with the length of the corridor. We observe the counterintuitive scaling mentioned by Shojaee et al. (2025) with models declining the request to do very long reasoning to solve complex problems. We report the reasoning effort increasing almost linearly with problem complexity for correct answers.

In the linear regression analysis, however only a small fraction of the variance is explained due to the noise of the measurements. This trend is observed only in the region where $\ell < 50$, since for larger corridor's lengths the number of correct predictions decreases significantly for all tested models. Main parameters of the linear regression fit are collected in Table 2.

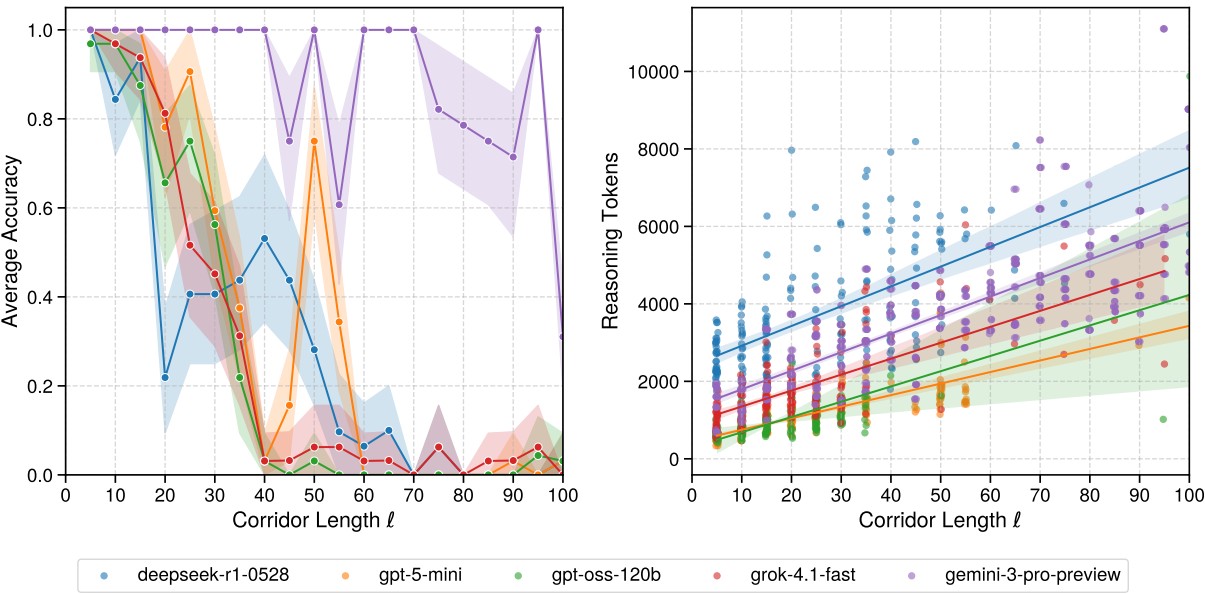

Figure 3: Accuracy and number of reasoning tokens for LRM experiment. Left: average accuracy. Error bars are computed as the 5th and 95th percentile of responses. Right: scaling behaviour of reasoning length against corridor length filtered for correct solutions only.

| Model | Slope | $R^2$ |
|---|---|---|
| deepseek-r1-0528 | 51.0879 | 0.345 |
| gemini3-pro | 47.899 | 0.622 |
| gpt-5-mini | 29.819 | 0.619 |
| gpt-oss-120b | 39.391 | 0.404 |
| grok-4.1-fast | 41.142 | 0.528 |

**Correct Answers**

| Model | Slope | $R^2$ |
|---|---|---|
| deepseek-r1-0528 | 86.321 | 0.247 |
| gemini3-pro | 53.746 | 0.3652 |
| gpt-5-mini | 55.154 | 0.138 |
| gpt-oss-120b | 85.864 | 0.124 |
| grok-4.1-fast | 989.393 | 0.192 |

**Wrong Answers**

Table 2: Fit parameters associated to the linear regressions performed on Figure 4.

In Figure 4 we further analyze the number of emitted tokens as a function of the corridor length parameter $\ell$, considering both correct and incorrect answers. Unlike Shojaee et al. (2025) which observed a counterintuitive reduction in the reasoning effort for problems above a certain threshold of difficulty, we observe a steady increase in the number of output tokens. What we found shows that the difficulty of a problem is not characterized by the decrease in the reasoning effort, but instead by the substantially higher variability in token counts of incorrect answers compared to correct ones. This suggests that when the model diverges from the correct reasoning trajectory, it can fail in multiple ways, whereas successful completions remain more concise and consistent, likely an effect of inductive bias of Group Reinforcement Policy Optimization (GRPO) post-training, where concise reasoning traces are preferred to lengthy ones (Sui et al., 2025). To

quantify this effect, we fit a robust regression of completion tokens against corridor's length for each model. Both slope and intercept appear model-specific: more efficient models, such as `gpt-5-mini`, show lower slopes and reduced variability across both correct and incorrect responses. Another relevant distinction can be made, highlighting differences in models' calibration. `deepseek-r1-0528` and `gpt-5-mini` display similar slopes and intercepts between correct and incorrect predictions, `gpt-oss-120b` instead reflect large differences in the regression parameters. A recurrent behavior is that for longer corridors, LRMs often reach the maximum allowed number of output tokens. We describe the main failure modes of reasoning in the next section. Briefly, the main motivation of failure are context overload resignation and counting errors that misrepresent the actual state of the game. We report the reasoning traces for the interested user at https://github.com/CarloNicolini/sokoban_benchmark_reasoning.

This repetitive looping behavior exemplifies what Lu et al. (Lu et al., 2025) classify as *unnecessary exploration* and failure to maintain a visited-state set. In a systematic search, an agent would track which configurations (or reasoning states) have already been explored and avoid revisiting them. The token-limit exhaustion we observe suggests that LRMs lack such memory: they repeatedly propose the same moves or reasoning steps without recognizing the cycle. This is evidence of *wandering* rather than systematic planning: the model explores aimlessly rather than pruning redundant paths. In a corridor setting, where the state space is essentially linear, even a simple mental tape of visited positions would suffice to prevent loops; the inability to maintain it indicates incapacity to maintain structured state tracking.

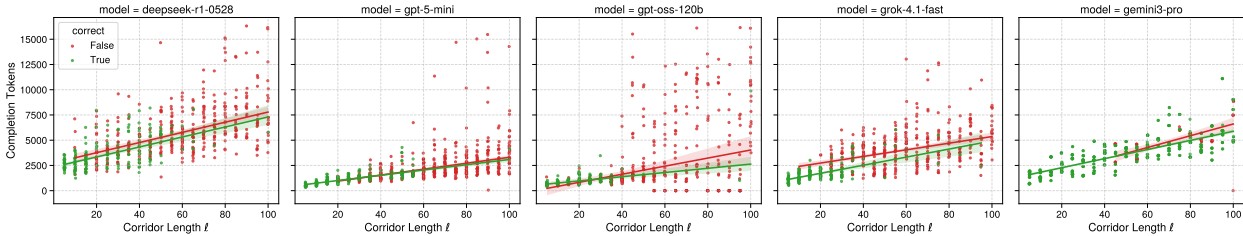

Figure 4: Number of completion tokens produced by each model as a function of corridor length. Separate linear regressions are fitted for correct and incorrect responses, with outliers excluded. A small jitter is added to the x-axis to improve visualization.

In Figure 5 we analyze the data from the point of view of prefix accuracy and Manhattan distance. The metrics show a decreasing trend for all models that is similar to that represented in Figure 3. Some patterns, like the peak at $\ell = 50$ for `gpt-5-mini` and the increase in accuracy around the central region for `deepseek-r1-0528`, are further accentuated, while prefix accuracy mostly reflects the exact accuracy in the case of the best performer `gemini-3-pro`. These observations highlight that the main source of errors in most Sokoban problems is related to counting mistakes. In terms of Manhattan distance, the optimal solution would have distance one as the player and the goal are separated by the box. Still `gemini-3-pro` has less problems than other models in keeping track of the current state, but with occasional yet present deviations. However, as observed sometimes by analysis and classification of the reasoning traces the player is positioned exactly on the goal, thus ignoring the spatial constraints of the problem resulting in a *off-by-one drift from the solution*.

These violations, where the predicted sequence places the player on the goal position despite walls and box, are instances of what Lu et al. (2025) terms *invalid exploration*. In a valid state-transition graph, certain moves (e.g., walking through walls, teleporting over boxes) are inadmissible. When a model proposes such transitions, it demonstrates that its internal representation does not faithfully track the game's physics and its constraints. LLMs hallucinate states unreachable under the true transition rules, producing reasoning traces that are syntactically plausible but structurally incoherent for the problem. The fact that even advanced reasoning models exhibit these errors underscores a core limitation: without explicit state-transition verification, test-time scaling cannot guarantee adherence to problem constraints and rules.

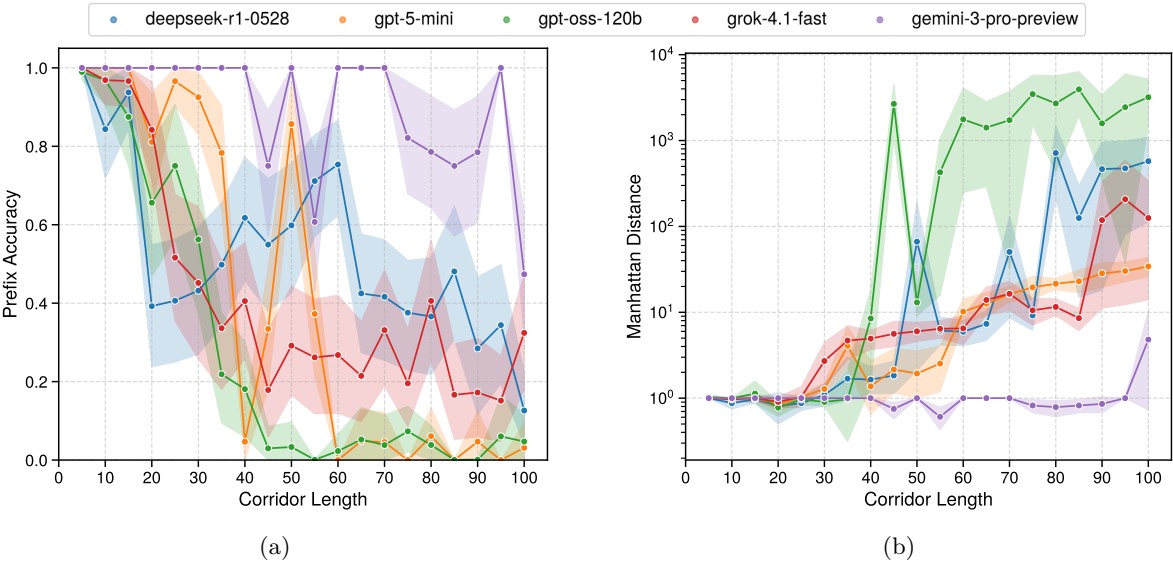

Figure 5: Other useful metrics represented as functions of the corridor's length. **Panel (a)** represents prefix accuracy, computed as described in Equation 2. **Panel (b)** represents Manhattan distance, computed as in Equation 3. Models' colors are the same as in Figure 3.

## 4.2 Failure modes analysis

We have further analyzed the thinking traces of the five models used throughout the experiments to understand what could be the main motivations driving the failure in solving the corridors. Our main hypothesis is that most of the failures are driven by the incorrect counting of the map size or an inability to sustain the reasoning over long sequence lengths. To validate this, we first constructed a classification schema by manually inspecting a representative subsample of the reasoning traces. From this qualitative analysis, we defined six recurrent failure modes. We then scaled the annotation process to more than 3,000 traces using OpenAI's `gpt-4.1` in an *LLM-as-judge* setup. The resulting schema comprises six primary categories: (i) *context overload resignation*, where the model becomes overwhelmed by the size or repetitiveness of the map and consequently abandons exact computation, resorts to blind guessing, or emits a placeholder; (ii) *counting error*, where the model correctly parses the map and mechanics but miscounts the required number of steps or empty spaces; (iii) *map hallucination*, where the model fabricates details about the ASCII layout that contradict the prompt; (iv) *rule misinterpretation*, involving a misunderstanding of Sokoban physics, such as implicitly allowing the agent to pull a box; (v) *logical inconsistency*, where the reasoning trace contains explicit internal contradictions or impossible state transitions; and (vi) *formatting error*, where the underlying reasoning is correct but the final answer violates strict syntactic constraints.

Table 3 clarifies the distribution of the observed failures across the evaluated models. First, *context overload resignation* and *counting errors* overwhelmingly dominate across all models, accounting for the vast majority of errors in every system. Context overload is particularly pronounced for `gpt-oss-120b` (52.5%) and `grok-4.1-fast` (43.2%). Importantly, we have observed that context overload tends to become the dominant pattern for longer corridors. The consistency of this pattern across architectures suggests that a primary bottleneck is not explicit rule violation, but rather an inability to reliably sustain focus and effort over longer horizons, frequently resulting in the model giving up.

Second, *counting errors* appear as the fundamental hurdle for the strongest solvers. For the top two performing models in our benchmark, `gemini-3-pro` and `gpt-5-mini`, pure counting errors constitute 44.1% and a massive 70.1% of their failures, respectively. This supports the interpretation that these advanced models rarely misunderstand the task or the environment; instead, they suffer from fragile positional bookkeeping and step counting. Their reasoning is locally coherent, but they miscalculate exact distances in large text grids.

Third, fundamental breakdowns in game logic—such as *rule misinterpretation* and *logical inconsistency*—are relatively rare overall. The notable exception is `deepseek-r1-0528`, which struggles more significantly with rule misinterpretation (13.3%) than the other models. Meanwhile, *map hallucinations* are relatively contained but remain a systematic issue for `grok-4.1-fast` (18.1%). Formatting errors are negligible for most models, though `gpt-oss-120b` exhibits a higher failure rate (11.0%) on syntax constraints alone.

Taken together, the evidence points to horizon-length sensitivity and cumulative arithmetic/counting errors as the dominant bottlenecks. The overall scarcity of rule misinterpretations and logical inconsistencies indicates that explicit misunderstandings of Sokoban dynamics are not the primary drivers of error. Instead, the prevalence of context resignation and counting errors highlights that maintaining exact spatial tracking and commitment to a long, structurally simple plan remains a nontrivial challenge even for the most recent reasoning-oriented models.

| **Model** | Context Overload | Counting Error | Map Hallucination | Rule Misinterpretation | Logical Inconsistency | Format Error | Other |
|---|---|---|---|---|---|---|---|
| `deepseek-r1-0528` | **37.7%** | 29.9% | 9.4% | 13.3% | 6.2% | 3.4% | 0.3% |
| `gemini-3-pro` | 40.5% | **44.1%** | 11.9% | 2.4% | - | 1.2% | - |
| `gpt-5-mini` | 17.0% | **70.1%** | 7.5% | 0.4% | 3.7% | 1.2% | - |
| `gpt-oss-120b` | **52.5%** | 32.9% | 0.7% | 2.7% | 0.3% | 11.0% | - |
| `grok-4.1-fast` | **43.2%** | 30.8% | 18.1% | 0.7% | 5.9% | - | 1.3% |

Table 3: Percentage distribution of failure modes in reasoning traces by model. Missing values (`nan`) in the table are represented as hyphens.

## 4.3 LRM-Modulo

Figure 6 shows the main results of the LRM-Modulo approach based on classical PDDL planning tools (domain, problem parsers and a problem solver). This configuration is introduced as a diagnostic setting rather than as a budget-matched comparison with reasoning-only LRMs. The objective is to disentangle failures due to plan generation from those due to execution and verification, by introducing an external symbolic solver as a controlled validation component. Unfortunately, preliminary experiments showed that not many models are both affordable in terms of costs and effective in tool-use tasks. Typical failures we encountered in testing models like `deepseek-r1-0528`, or others not shown here as `gemini-2.5-flash`, or `haiku-3.5` include limited capability to interact with tools, difficulty in generating coherent PDDL problems even for simpler Sokoban instances, and inability to stop calling tools after a given number of attempts. `gpt-5-mini` and `gemini-3-pro` were the only models among those tested that consistently generated accurate PDDL problems and interacted with the solver while respecting the prompt constraints. Due to the higher costs of the experiments in the LRM-Modulo setting we limited the experiment's repetitions per corridor rotation $n_t$ to four. Nonetheless, `gpt-5-mini` and `gemini-3-pro` exhibit high stability in accuracy and in the number of reasoning tokens, allowing a valid evaluation even with a reduced number of repetitions.

In Figure 6a, the absence of sharp peaks and the slower descending trend highlights a more regular accuracy behavior compared to that shown in Figure 3 for LRMs alone. However, a higher variability for both models is observed, mainly due to non-homogeneous performances across experimental trials and map rotations for a fixed corridor length. Visual inspection of the results reveals a significant imbalance between accuracy in vertical and horizontal corridors (Figure 8), showing that also in LRM-modulo settings models struggle to solve vertical corridors. At the same time, a detailed analysis of the source of these errors indicates two main causes of failure. One occurs when there are syntax errors in the generated PDDL problems, producing error messages when calling the solver tool. The other occurs when generated PDDL problems are syntactically correct but do not faithfully represent the actual Sokoban configuration. In our data, first-type errors occur only 7 times out of all four trials of the 80 corridor configurations, meaning that in the large majority of cases the solver compiles correctly and produces a valid plan. The charts depicting the prefix accuracy and the Manhattan distance, represented in Figure 7, confirm that in many cases the generated PDDL representation leads to solutions in which the player, although moving in the correct direction, does not execute the number of moves required to push the box to the goal position.

By utilizing an expert-validated PDDL domain and a solver strictly governed by logical constraints, we have effectively eliminated the risk of invalid transitions. In this assisted regime, improvements should not be interpreted as enhanced intrinsic planning ability, but rather as the effect of external validation and iterative reformulation. Even under this upper-bound configuration, performance remains unstable and often requires multiple retries, indicating that the core difficulty lies in maintaining a consistent internal representation of the spatial environment. While we observe that for longer corridors `gpt-5-mini` is improving accuracy with respect to the one shot case, on the stronger `gemini-3-pro` model, the LRM-Modulo results are almost consistently worse than the one-shot case. Evidence suggests this difficulty may stem from a systematic difficulty in correctly quantifying the map length and, consequently, in generating the exact sequence of actions required to solve the corridor.

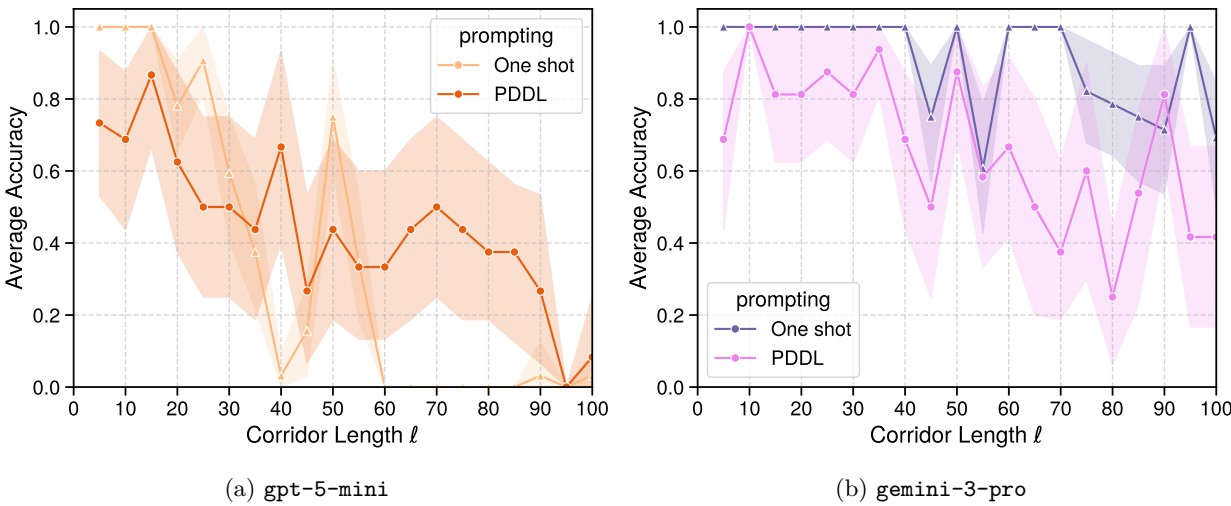

(a) `gpt-5-mini`          (b) `gemini-3-pro`

Figure 6: Average accuracy of the LRM-Modulo approach for `gpt-5-mini` and `gemini-3-pro` compared with their relative one-shot case. In one shot inference, model colors are maintained as in the previous figures. Error ribbons are computed as 5 and 95 percentiles.

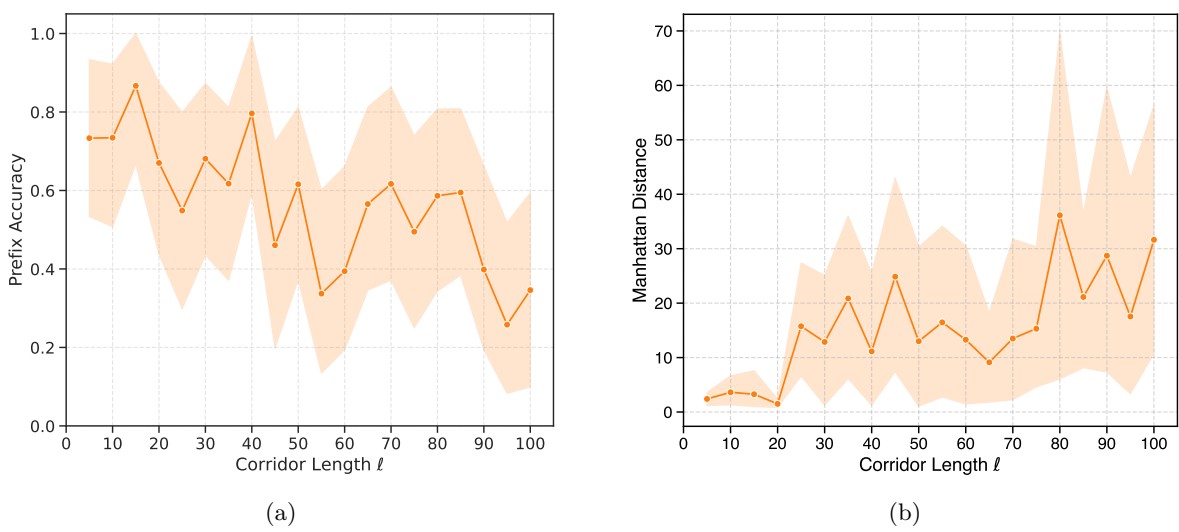

(a)          (b)

Figure 7: Additional metrics for `gpt-5-mini` in the LRM-modulo framework. **Panel (a)** represents prefix accuracy (Eq. 2). **Panel (b)** shows Manhattan distance (Eq. 3).

# 5 Conclusions

The assessment of the long-horizon planning capacities of LRMs is both required and attainable. Adhering to the principle of beginning with simplistic settings before advancing to more intricate ones, we propose utilizing a simplified version of Sokoban as a controlled environment to evaluate planning capabilities. Our observations, in agreement with prior research, suggest that long planning abilities of LRMs may not only be related to problem complexity but from lack of more elementary initial abilities like counting.

We observe that even advanced reasoning models struggle to solve Sokoban instances that require anticipating the goal state more than 25–30 moves ahead. We discussed several possible causes for this limited performance in the limitations section, including the absence of textual cues and the inability to reliably store intermediate states within model hidden representations.

The use of proprietary models hinders us to understand potential training exposure. However we notice that the set of maps we have used is highly atypical and unlikely to appear in natural Sokoban corpora due to their trivial gameplay structure. While contamination cannot be fully excluded, the consistent horizon-dependent degradation across multiple frontier models suggests that memorization alone does not explain the observed behaviour.

Equipping language models with a PDDL parser, validator, and solver slightly improves planning capabilities on average, but not enough to overcome the lack of inherent spatial grounding. We found that the basic, initial inability to track counts remains a persistent bottleneck. This issue surfaces even in LRM modulo settings where external symbolic engines are used, proving that offloading logic to a solver cannot fully fix a model that cannot faithfully represent space and constraints.

More broadly, our observations align with recent characterizations of reasoning models as "wanderers" rather than systematic explorers: linear corridors exemplify a setting where minimal branching but substantial depth exposes how small per-step errors in state tracking (counting drift, visited-state amnesia, invalid transitions) compound exponentially. Consequently, test-time scaling alone cannot overcome these structural limitations without architectural innovations, short horizon error tracking or explicit symbolic grounding.

## 5.1 Current limitations and future work

Our study is intentionally narrow; here we outline the main constraints and limitations. We focus on one-box linear corridors, which test long-horizon counting and state maintenance rather than the full difficulty of multi-box Sokoban with deadlocks. Thus, the benchmark provides only a lower bound on planning ability. Moreover we use only *solvable* maps, where the optimal solution exists and has demonstrably low branching factor. We also specify in the system prompt that the LRM is faced with a solvable map. We believe that posing the LRM unsolvable maps with similar corridor-based layout could highlight further reasoning limits. A simple but interesting possibility to have impossible maps would be to switch the player position with the box, thus leaving the player between the box and the goal with no possibility to push the box toward the goal. We leave the study of unsolvable maps for a future work. For evaluation, we use exact-plan validation against a reference generator. Although this is stricter than necessary in general Sokoban, where multiple optimal plans may exist, it is suitable for corridors; future work will instead use solver-based verification to handle maps with multiple valid solutions. We also find sensitivity to prompt formatting, especially orientation-related effects such as the many newlines in vertical maps. Alternative encodings, such as row/column numbered grids or other textual cues, may reduce this issue. Another variability source is model metadata and provider backends: although all calls go through one routing layer, backend implementations and model revisions can change over time. We log identifiers and dates, but some instability is inherent in API-based evaluations. Pretraining contamination is another concern; corridor rotations lower the chance that specific plans were memorized but do not eliminate it. Finally, corridor tasks have limited external validity, since success or failure may not transfer to richer planning domains. We treat these settings mainly as a sanity check, with follow-up experiments planned to add obstacles, branching structures, and deadlocks.

## Societal Impact

This paper presents work whose goal is to advance the field of Machine Learning through clearer diagnostics of long-horizon planning. While any benchmark could have indirect downstream effects by steering research agendas, we do not identify specific societal risks unique to this work beyond standard concerns about evaluation misuse. We therefore do not highlight any particular societal impacts at this time.

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

## Appendix

## A  Prompts for 1-shot Inference Settings

In this section we report the detailed prompts that we have used throughout our experiments with reasoning models alone. Importantly, in this case we don't try to elicit chain of thought. A simple solved problem is provided as the 1-shot example.

---

**System Prompt**

```
You are an assistant that helps in solving assigned Sokoban games.
Your task is to examine the provided Sokoban problem and find a solution.
All provided Sokoban problems are assigned in form of ASCII maps.
The mapping is the following:

'''
    @ - Player
    + - Player on Goal
    $ - Box
    * - Box on Goal
    . - Goal (Empty)
    # - Wall Brick
'''

The game is solved when the box is pushed into the goal position, hence when the position of '$' coincides with the position of '.'.
The player can move in all the empty spaces of the ASCII map while respecting the walls.
When the player is adjacent to a box, the player can push the box into an adjacent empty space.
After pushing a box, the new position of the agent will be the position of the box before the push.
The player cannot pull the box, only push it.
The actions you can perform in the game are:

'''
    L - Move Left
    R - Move Right
    U - Move Up
    D - Move Down
'''

All provided problems CAN be solved.
You must give your solution in form of a sequence of allowed actions, separated by commas.
You must give only the sequence of actions, without any additional text or explanation.
You must enclose your solution inside the tags <plan> </plan>.

The following is an example of a Sokoban problem and its solution:

Problem:

#####
#@  ##
## $ ##
 #   #
 ##. ##
  ## #
   ###

Solution:

<plan>
R,R,D,D
</plan>
```

---

**User Prompt**

```
Here is the Sokoban problem to solve, enclosed in triple backtics:

'''
{{ sokoban_map }}
'''
```

# B  Prompts for LLM-Modulo Settings

In this section, we show the system prompt we used for the experiments on LLM-Modulo settings. The user prompt remains the same as shown in Appendix A. The system prompt includes the human-designed PDDL domain of a typical Sokoban game[3].

## B.1  System Prompt

Here are the system prompts and the PDDL domain being used for the experiments in LLM-Modulo settings. The model is just required to generate the PDDL problem to be sent to the solver.

---

[3]https://verificationglasses.wordpress.com/2021/01/02/sokoban-pddl

## System Prompt

```
You are an assistant that helps in solving assigned Sokoban games.
All provided Sokoban problems are assigned in form of ASCII maps and CAN be solved.
The mapping is the following:

\'\'\'
    @ --- Player
    + --- Player on Goal
    \$ --- Box
    * --- Box on Goal
    . --- Goal (Empty)
    \# --- Wall Brick
\'\'\'

Given the PDDL domain of a generic Sokoban game, your task is to generate a valid PDDL problem representation of the provided ASCII Sokoban
    problem.
Once you generate the PDDL problem, your final goal is to find a plan that solves the problem.
You have access to a set of tools to help you achieve your goal.
Always use the solve\_problem tool to solve the problem, do not try to solve it yourself.
If you encur in any error while solving a problem with the tool, try to fix it and call the tool again.
Retry up to 3 times at maximum if needed.

Here is the PDDL Sokoban domain, enclosed in triple backtics:
\'\'\'
\{\{PDDL\_domain\}\}
\'\'\'

IMPORTANT:
Your final answer must contain both the the PDDL problem and the solution to the problem without any additional text or explanation.
You must separately enclose the PDDL problem inside the tags <problem> </problem>, and the solution inside the tags <plan> </plan>.
If, after the third attempt, you are unable to get a solution from the solver, provide the error message you received from the tool inside
    the <plan> </plan> tags.
If at the end of your process the solve\_problem tool gets called without errors and returns a solution, write <solver>True</solver>,
    otherwise write <solver>False</solver>.

Example output:
\'\'\'yaml
problem: <problem>PDDL problem here</problem>
plan: <plan>PDDL plan from solver here</plan>
solver: <solver>Boolean checking whether solve\_problem tool was called successfully</solver>
\'\'\'
```

## B.2 PDDL Domain

Here the human authored PDDL domain used in the above system prompt is reported for completeness.

## PDDL Domain

```
(define (domain sokoban)
    (:predicates (wall ?x ?y) (box ?x ?y) (at ?x ?y) (inc ?p ?pp) (dec ?pp ?p))
    (:action move-up
        :parameters (?x ?y ?xn)
        :precondition (and (at ?x ?y) (not (wall ?xn ?y)) (not (box ?xn ?y)) (dec ?x ?xn))
        :effect (and (not (at ?x ?y)) (at ?xn ?y))
    )
    (:action move-down
        :parameters (?x ?y ?xn)
        :precondition (and (at ?x ?y) (not (wall ?xn ?y)) (not (box ?xn ?y)) (inc ?x ?xn))
        :effect (and (not (at ?x ?y)) (at ?xn ?y))
    )
    (:action move-right
        :parameters (?x ?y ?yn)
        :precondition (and (at ?x ?y) (not (wall ?x ?yn)) (not (box ?x ?yn)) (inc ?y ?yn))
        :effect (and (not (at ?x ?y)) (at ?x ?yn))
    )
    (:action move-left
        :parameters (?x ?y ?yn)
        :precondition (and (at ?x ?y) (not (wall ?x ?yn)) (not (box ?x ?yn)) (dec ?y ?yn))
        :effect (and (not (at ?x ?y)) (at ?x ?yn))
    )
    (:action push-up
        :parameters (?x ?y ?xn ?xnn)
        :precondition (and (at ?x ?y) (not (wall ?xn ?y)) (box ?xn ?y) (dec ?x ?xn) (not (wall ?xnn ?y)) (not (box ?xnn ?y)) (dec ?xn ?xnn))
        :effect (and (not (at ?x ?y)) (at ?xn ?y) (not (box ?xn ?y)) (box ?xnn ?y))
    )
    (:action push-down
        :parameters (?x ?y ?xn ?xnn)
        :precondition (and (at ?x ?y) (not (wall ?xn ?y)) (box ?xn ?y) (inc ?x ?xn) (not (wall ?xnn ?y)) (not (box ?xnn ?y)) (inc ?xn ?xnn))
        :effect (and (not (at ?x ?y)) (at ?xn ?y) (not (box ?xn ?y)) (box ?xnn ?y))
    )
    (:action push-right
        :parameters (?x ?y ?yn ?ynn)
        :precondition (and (at ?x ?y) (not (wall ?x ?yn)) (box ?x ?yn) (inc ?y ?yn) (not (wall ?x ?ynn)) (not (box ?x ?ynn)) (inc ?yn ?ynn))
        :effect (and (not (at ?x ?y)) (at ?x ?yn) (not (box ?x ?yn)) (box ?x ?ynn))
    )
    (:action push-left
        :parameters (?x ?y ?yn ?ynn)
        :precondition (and (at ?x ?y) (not (wall ?x ?yn)) (box ?x ?yn) (dec ?y ?yn) (not (wall ?x ?ynn)) (not (box ?x ?ynn)) (dec ?yn ?ynn))
        :effect (and (not (at ?x ?y)) (at ?x ?yn) (not (box ?x ?yn)) (box ?x ?ynn))
    )
)
```

# C  LRM-Modulo breakdown by map rotations

In this section we disaggregated the LLM-modulo results by map rotation, reporting accuracies separately for all rotations. For clarity, we focus on the two leading models in the overall evaluation, `gpt-5-mini` and `gemini-3-pro`. Accuracies are averaged over the four experimental trials for each configuration.

Figure 8 shows that performance is not rotation-invariant for either model. While both systems maintain relatively strong accuracy on 0° and 180° layouts, and exhibit moderate degradation at 90°, the 270° rotation consistently produces the worst results.

This pattern is consistent across both models despite architectural differences. The degradation at 270° suggests a systematic sensitivity to orientation rather than simple variance. Since the underlying problem is geometrically equivalent under rotation, the asymmetry indicates that the models do not encode a fully rotation-equivariant representation of the grid. The 270° case *likely* compounds directional priors with accumulated state-tracking error, leading to unstable representation of the map.

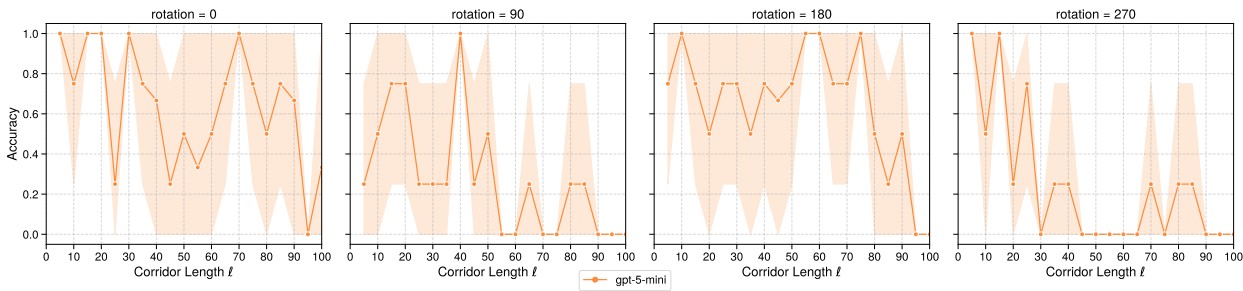

(a) **gpt-5-mini** — Accuracies in LLM-Modulo settings for rotations 0°, 90°, 180°, 270°.

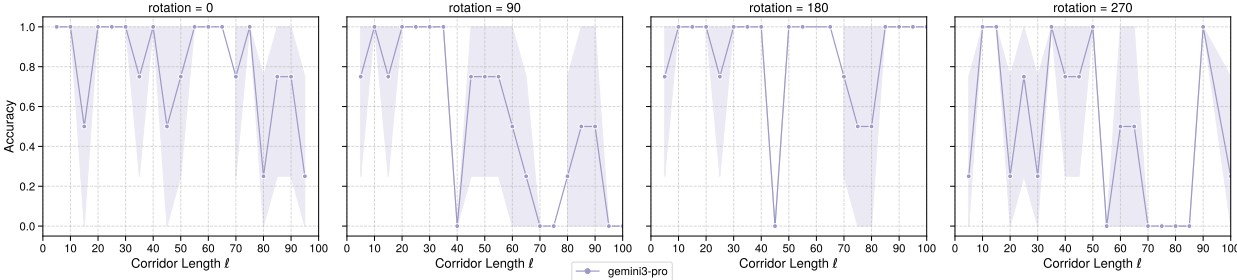

(b) **gemini-3-pro** — Accuracies in LLM-Modulo settings for rotations 0°, 90°, 180°, 270°.

Figure 8: Accuracies in the LLM-modulo setting, averaged over four experiment trials for each Sokoban corridor rotation. Error bars denote variability across trials.

# D  Thinking traces' classifier

Here we provide the full prompt that we have used to instruct `gpt-4.1` to analyze and classify the thinking traces produced by our experiments, so to classify the reasoning failures into the six modes discussed in section 4.2.

**System Prompt**

```
You are an expert evaluator of reasoning traces generated by large language models on sequential planning tasks (specifically Sokoban-style
     grid puzzles).
Your task is to analyze a single reasoning trace provided by the user and classify the dominant failure mode exhibited by the model.
You must output strictly valid JSON and nothing else.

# Task Definition
The user will provide a reasoning trace as a raw string. The trace represents the step-by-step internal reasoning or action sequence of a
     model attempting to solve a deterministic planning puzzle.
```

```
Your job is to determine which of the following failure categories best describes the trace:

- map_hallucination
- counting_error
- rule_misinterpretation
- logical_inconsistency
- context_overload_resignation
- formatting_error
- other

You must choose exactly one label.
If multiple issues appear, select the dominant failure that most directly explains the final failure.

# Output Format (MANDATORY)

You must output exactly this JSON schema:

{
  "label": "<one_of_the_allowed_labels>",
  "confidence": <float_between_0_and_1>,
  "evidence": "<brief concrete justification>"
}

## Rules:
- Output must be valid JSON.
- Do not include any text before or after the JSON.
- Do not include markdown formatting (like ```json) in the final output string.
- Confidence must be a decimal between 0 and 1.
- Evidence must reference specific observable patterns or quotes in the trace.
- Label must be exactly one of the seven allowed strings.

## Label Definitions (Operational Criteria)

Use the following precise definitions to classify the trace.

### `map_hallucination`
Definition: The model makes incorrect assumptions or fabricates details about the map's layout, dimensions, or the relative positions of
    elements that contradict the provided ASCII representation.
Typical signals:
- Claiming the map has branches when it is a single straight corridor.
- Misidentifying the starting coordinates of the player, box, or goal right from the beginning.
- Asserting the presence of obstacles that do not exist in the prompt.

### `counting_error`
Definition: The model correctly understands the map and rules but fails to accurately count the number of grid cells, empty spaces, or
    required moves to navigate the map.
Typical signals:
- Incorrectly calculating the distance between the player and the box.
- Outputting 51 'D' moves when 53 were required.
- Arithmetic errors in reasoning (e.g., "100 - 49 = 50").
- Near-miss or off-by-one drift errors.

### `rule_misinterpretation`
Definition: The model misunderstands or misapplies the mechanical rules of the game (e.g., Sokoban physics).
Typical signals:
- Assuming the player can pull a box.
- Forgetting that the player ends up in the box's previous position after a push.
- Attempting to push a box into a wall or through another object.
- Stating the player must "go around" the box in a 1D corridor (which is impossible).

### `logical_inconsistency`
Definition: The reasoning contains explicit contradictions or impossible state transitions relative to its own previous deductions.
Typical signals:
- The model asserts a fact, then contradicts it a few sentences later (e.g., "The box is at row 50... therefore the box is at row 42").
- Teleportation of the player or box in the model's internal mental state.
- Position updates that do not logically follow from the previous step's math.

### `context_overload_resignation`
Definition: The model becomes overwhelmed by the size, repetitiveness, or length of the input map, leading it to abandon exact calculations,
    guess blindly, or output a placeholder.
Typical signals:
- Explicit complaints about the length (e.g., "This is insane", "Obviously the map is huge").
- Giving up on counting and guessing a round number (e.g., "I'll just output 100 U's").
- Outputting an empty or artificially truncated plan because it is "too tedious to write".

### `formatting_error`
Definition: The model solves the puzzle correctly in its reasoning but fails to follow strict output formatting instructions.
Typical signals:
- Failing to comma-separate the actions (e.g., outputting "DDDD" instead of "D,D,D,D").
- Forgetting to enclose the final answer in the required `<plan>` tags.
- Adding conversational filler inside the final output block where only a sequence was requested.

### `other`
Definition: A failure mode that does not neatly fit into any of the above categories (e.g., generating an optimal path that is valid, but the
    system classifies it as a failure for a different, unknown reason).

# Tie-Breaking Rule

If multiple failure patterns appear, use this hierarchy to determine the root cause:
1. Prefer `context_overload_resignation` if the model explicitly gives up or guesses due to length.
2. Otherwise prefer `map_hallucination` if the model's foundational understanding of the text-based map is completely wrong.
3. Otherwise prefer `rule_misinterpretation` if the model's physics/mechanics logic is flawed.
4. Otherwise prefer `logical_inconsistency` if the model contradicts its own internal state tracking.
5. Otherwise prefer `counting_error` if the logic is sound but the math/counting is simply wrong.
6. Otherwise use `formatting_error` if the plan is factually correct but fails syntax constraints.

## Style Constraints
- Be conservative and evidence-driven.
- Do not speculate beyond the trace.
- Base decisions only on observable text patterns.
- Keep evidence concise but highly specific (quote numbers, coordinates, or exact phrases).

# What Each Failure Looks Like in Practice
```

```
Below is the diagnostic intuition you should expect when inspecting failing traces.

## Context Overload & resignation
Expect to see the model hallucinate line numbers into the thousands, start skipping blocks of text, or output phrases like "I will guess 202
        U moves" or "Given time constraints, I will output a placeholder." The model recognizes the task but refuses to do the hard
        combinatorial work.

## Map Hallucination
Expect to see the model misread the raw ASCII string. For example, reading '#.   $   @#' and concluding the player is to the *left* of the
        box, or concluding the map is 12 rows tall when it is actually 50 rows tall.

## Rule Misinterpretation
Expect the model to correctly identify the map but fail at the game. For example: "The player pushes the box to row 5. Now the player is at
        row 7." (In Sokoban, pushing a box to row 5 from row 6 means the player *must* step into row 6, not stay at 7).

## Logical Inconsistency
Expect the model to lose track of its own narrative. "We need 46 moves down. Then 51 pushes down. Total: 85 moves." (46 + 51 is 97, not 85).
        Or, "Player is at row 20, moves up 5 times, is now at row 25." (Moving up should decrease the row number).

## Counting Error
Expect meticulous, step-by-step reasoning that is conceptually flawless but simply miscounts the number of spaces in a string of 40 empty
        spaces, resulting in an output like 39 'R's instead of 40 'R's.

## Formatting Error
Expect a perfect count and perfect logic, but the final output is '<plan>D D D</plan>' instead of '<plan>D,D,D</plan>', or trailing
        commas '<plan>D,D,D,</plan>'.
```

# E   Chain of thought prompting

We have studied the effect of explicitly eliciting chain-of-thought reasoning through the instruction "*think step by step*", added to the original prompt used for LRM models. We evaluate two recent frontier systems with different design philosophies: `gpt-4.1` (OpenAI, 2025a), a high-capacity but non-reasoning model, and `gemini-3-pro` (Google DeepMind, 2025), a model explicitly optimized for reasoning. Results are shown in Figure 9.

For `gpt-4.1`, adding chain-of-thought produces a small but consistent improvement. The gain is most visible at intermediate corridor lengths, where single-shot heuristics fail. This suggests that, for a non-reasoning architecture, prompting stepwise verbalization partially stabilizes internal state tracking and reduces premature answer emission. However, the improvement remains limited: performance still degrades with length, indicating that explicit verbal reasoning does not fundamentally extend its effective planning horizon.

In contrast, `gemini-3-pro` shows a slight degradation when forced to "think step by step". Since this model already incorporates an internal reasoning mechanism, externally elicited chain-of-thought may interfere with its optimized inference procedure. The added verbosity can introduce exposure to intermediate generation errors or disrupt compact latent reasoning traces. The result is that explicit chain-of-thought does not provide additional benefit and reduces accuracy.

These findings indicate that chain-of-thought prompting is architecture-sensitive. It can modestly assist non-reasoning models by encouraging structured intermediate computation, but it does not universally improve performance and may be counterproductive for models already designed to reason internally.

# F   Extending corridors' length

To evaluate the extrapolation limits of `gemini-3-pro`, we extended corridor lengths to 200 to determine if previous performance drops indicated a fixed capacity ceiling or a relative difficulty threshold. Figure 10 shows that the transition to low performance is merely displaced rather than eliminated. The qualitative degradation pattern remains consistent, suggesting that the model's planning horizon is expanded but actually finite.

Consequently, the model fails to demonstrate scale-invariant length generalization as it appears to extend its effective planning depth by a bounded margin. This indicates that increasing model scale shifts the failure point further out but does not fundamentally resolve the collapse that occurs once the planning horizon exceeds the model's learned threshold.

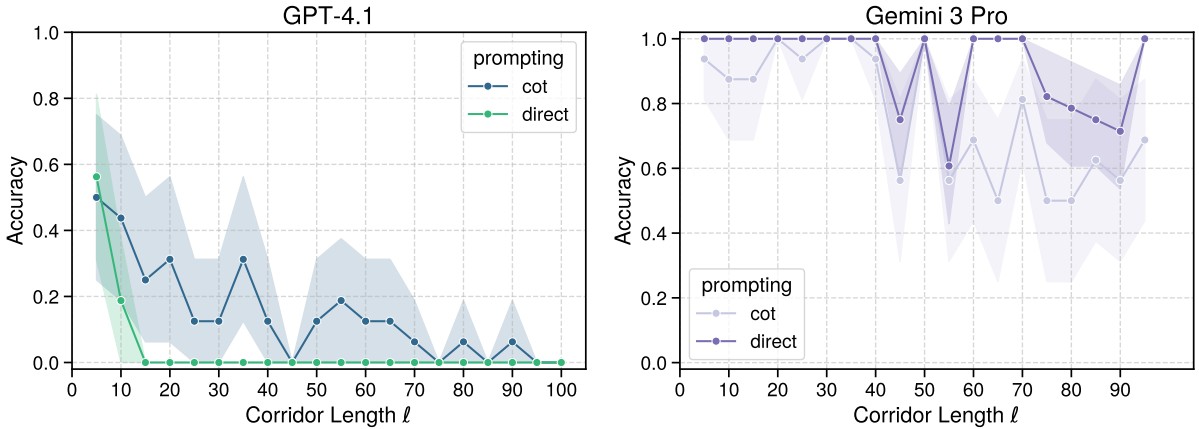

Figure 9: Accuracy on corridors from length 5 to 100 eliciting chain of thought reasoning on `gpt-4.1` and `gemini-3-pro`. Values are averaged over 4 rotations and 8 trials as in the other presented results.

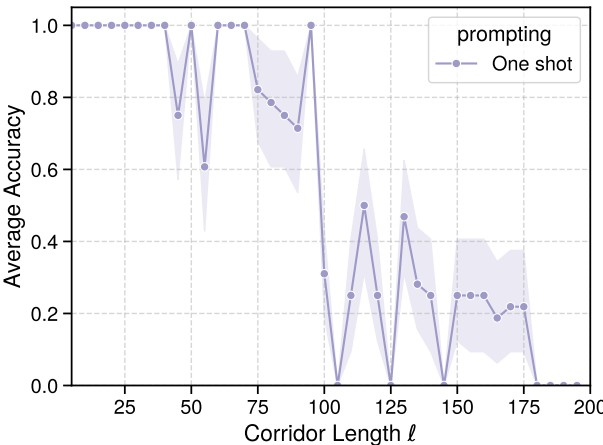

Figure 10: Full scale analysis of `gemini-3-pro` for all four rotations and eight trials for size in $[5, 200]$.

## G  Freetski: A highly simplified Klotski game

To further investigate the spatial reasoning and long-horizon planning capabilities of LRMs, we introduce a new diagnostic game called "*Freetski*". Freetski is a highly simplified variant of the popular sliding-block puzzle, Klotski[4]. Unlike standard Klotski, which requires coordinating multiple differently sized blocks to clear a path, Freetski restricts the environment to a single rigid block enclosed within a long corridor (`#`), similarly to our simplified Sokoban maps. The block is composed of multiple boxes (`$`), and the goal is to maneuver this block entirely out of the walls through a specific opening. In Figure 11 an example of dynamics is shown. In our experiments however we limited, as in the Sokoban case, to straight, long and narrow corridors without corners as in the figure. The dataset we used is freely available[5].

Any spatial translation (Left, Right, Up, or Down) must be applied synchronously to all of its constituent pieces because the block is a rigid body. The model is required to output a matrix of movement sequences, applying the exact same directional operations to every piece to maintain the object's shape. Hence, Freetski introduces an additional layer of spatial coordination: the model must track not only the distance to the goal but also the physical connectedness of the multi-piece block and the contact to the static walls.

---

[4] https://en.wikipedia.org/wiki/Klotski
[5] https://huggingface.co/datasets/Linello/freetski

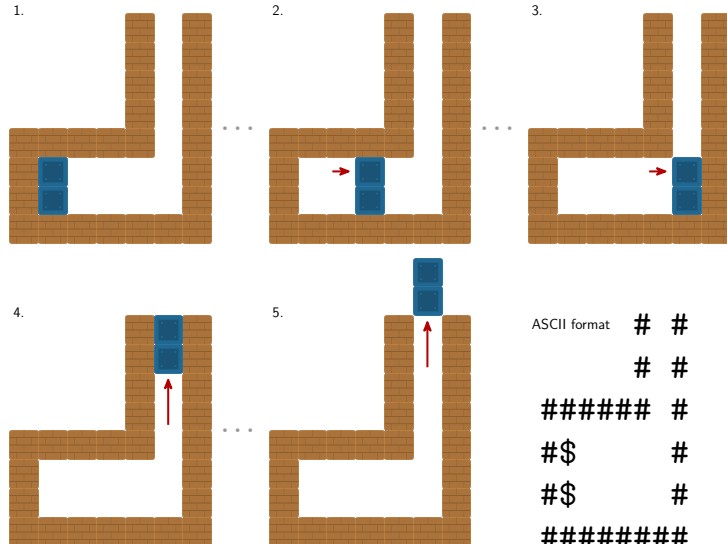

Figure 11: A sequential demonstration of the agent moving the $1 \times 2$ block toward the exit of the labyrinth. The sequence (A through E) illustrates the necessary column alignments to clear the vertical shaft. In the bottom right corner the representation of the labyrinth in ASCII.

As illustrated in Figure 12, performance on Freetski corridors for very small $2 \times 2$ blocks is highly volatile and characterized by a generally low average accuracy. Although the model occasionally manages to solve specific map lengths, the performance frequently collapses across the evaluated corridor lengths. The plot of reasoning tokens reveals a familiar linear upward trend as the solution length increases, but it is accompanied by extreme variance (ranging from roughly 500 to nearly 3000 tokens for similar map lengths). This behavior strongly mirrors the "wandering" phenomenon detailed in the main text: instead of systematic spatial planning, the model struggles with positional bookkeeping of the multi-piece object, getting stuck in loops or producing structurally incoherent outputs. Ultimately, this reinforces our conclusion that current LRMs struggle in maintaining reliable internal state representations over long reasoning horizons.

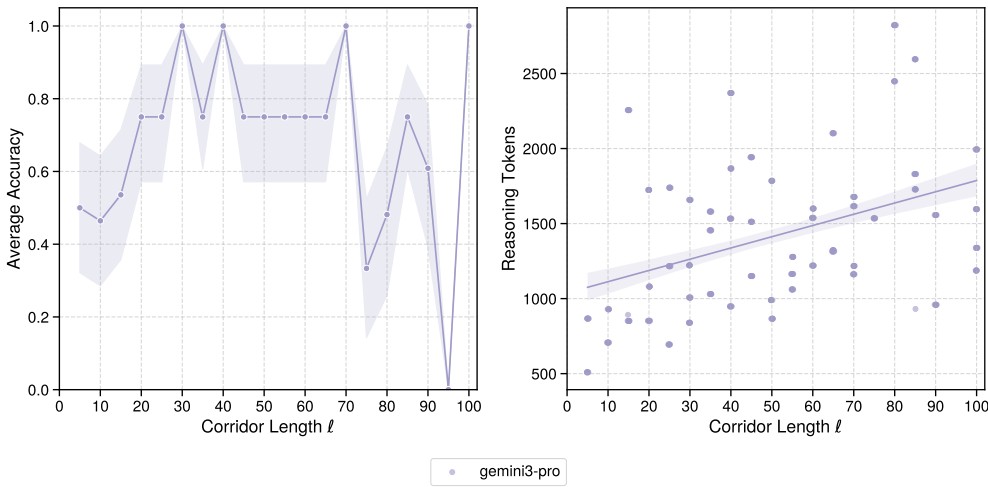

Figure 12: Average accuracy (left) and reasoning token usage (right) for the Freetski game as a function of corridor length $\ell$ using `gemini-3-pro`. The highly erratic accuracy and large variance in token consumption highlight the models' severe difficulties in coordinating multi-piece rigid body translations over long horizons.

