# OpenReview forum: "SokoBench: Evaluating Long-Horizon Planning and Reasoning in Large Language Models"
_TMLR — Accepted by TMLR_

### Review · Reviewer_Xvrr · 2026-02-16

**Summary Of Contributions:**

Negative empirical results on a 1-dimensional variant of the Sokoban puzzle (push box from start to target in a 2D grid), in both the 1-shot and tool use (with access to a PDDL planner) settings.

**Audience:**

Yes

**Audience Explanation:**

As the paper's related work section already states, there is prior work on evaluating LLMs on Sokoban puzzles [1]. This paper simplifies the puzzle so that the unique optimal solution is to push the box along a single cardinal direction (up/down/left/right) for some variable length. I suppose some people who wonder why LLMs can't solve the harder general Sokoban puzzles will be interested in knowing whether they can solve the simplified version, to guide further investigation in the general case.

But this paper's experimental results are in the negative (i.e. LLMs can't even solve the 1D version), and it doesn't provide deeper investigation, unlike the cited work [2] that investigates why LLMs faced difficulty in character-level tasks. Since [2] concluded that eliciting more reasoning helps character-level tasks, one would expect this paper to try that also, and see if it changes the results.


[1] Karthik Valmeekam, Kaya Stechly, Atharva Gundawar, and Subbarao Kambhampati. A systematic evaluation
of the planning and scheduling abilities of the reasoning model o1. Transactions on Machine Learning
Research, 2025. ISSN 2835-8856. URL https://openreview.net/forum?id=FkKBxp0FhR
[2] Nan Xu and Xuezhe Ma. Llm the genius paradox: A linguistic and math expert’s struggle with simple wordbased
counting problems. In Proceedings of the 2025 Conference of the Nations of the Americas Chapter of
the Association for Computational Linguistics: Human Language Technologies (Volume 1: Long Papers),
pp. 3344–3370, 2025.

**Claims And Evidence:**

No

**Claims Explanation:**

Overall, this is a purely empirical paper that reports the performance of some open and closed LLMs on a simplified 1-dimensional variant of the spatial push-box-to-target Sokoban puzzle. In so far as there are claims in the paper, most of them are experimental observations that are backed by measurements, so those are fine.

But there is one statement that is problematic when interpreted as a claim. The abstract says that the authors find:
> a consistent degradation in planning performance when more than 25 moves are required to reach the solution, **suggesting a
fundamental constraint on forward planning capacity**

A fundamental constraint should either be proved or conjectured (e.g. the classic impossibility proof that single-layer perceptron cannot represent the XOR function). By "suggesting", I can't tell whether the authors mean to make a claim or not. Also I don't see how it is possible for any empirical result on any problem instance to make one believe one way or another about whether there exists a fundamental constraint on a black-box model's ability in general. If this is a claim, then no amount of empirical evidence is sufficient; one should just proof it, or not even state it at all.

**Requested Changes:**

1. Be clear about whether the authors really want to claim a "fundamental constraint on forward planning capacity".
2. Try more strategies, like those in [2], to see if results improve.

[2] Nan Xu and Xuezhe Ma. Llm the genius paradox: A linguistic and math expert’s struggle with simple wordbased
counting problems. In Proceedings of the 2025 Conference of the Nations of the Americas Chapter of
the Association for Computational Linguistics: Human Language Technologies (Volume 1: Long Papers),
pp. 3344–3370, 2025.

---

> ### Author Response · Authors · 2026-03-04
> **Response to reviewer Xvrr**
>
> We appreciate the reviewer’s emphasis on epistemic precision and additional diagnostic depth.
>
> ### **1.Clarification of the *fundamental constraint* language**
>
> We agree that the original wording could be interpreted as overclaiming.
> We have carefully revised the abstract and main text to remove any implication of a formal impossibility result.
> The manuscript now explicitly frames the observed degradation as an empirical phenomenon under single-pass autoregressive decoding.
> We state clearly that our results do not establish universal limits of the model class.
> Now the abstract reports:
>
> > We show that equipping LRMs with Planning Domain Definition Language (PDDL) parsing, validation, and solving tools allows for modest improvements, suggesting that character level counting and long yet simple state tracking might not be overcome by test-time scaling approaches alone.
>
> ### **2. Additional elicitation strategies**
>
> The reviewer suggested exploring alternative prompting strategies.
> In the revised manuscript, we address this concern in two ways.
> First, following the suggestion, we expanded the empirical study to include two additional frontier reasoning systems, `gemini3-pro` and `grok-4.1-fast`.
> These models substantially extend the capability frontier relative to the originally evaluated systems.
> The new results confirm the central empirical pattern.
> `gemini3-pro` is the strongest performer overall but still exhibits clear degradation at moderately larger horizons: in the appendix we report the effect over expanded corridor lengths from $100$ to $200$.
> `grok-4.1-fast`, on the other hand, despite supporting extremely long contexts (up to two million tokens), shows near-zero accuracy already around $\\ell \approx 40$.
> This evidence indicates that the observed breakdown cannot be attributed purely to context window limitations.
>
> We have included the updated model accuracy calculations including the two above models in the revised manuscript (Figure 1).
> We have also included the additional test on `gemini3-pro` on longer corridors in the appendix of the revised manuscript to show that the performance still degrades to zero for longer corridors.
> Second, regarding explicit chain-of-thought prompting, we have run two experiments for testing the effect of chain of thought prompting both with `gemini-3-pro` (the best reasoning model used in our revised experiments) and with a non-reasoning model (`gpt-4.1`) that was not being used in our experiments, but it was useful as a comparison since it's a non-reasoning model that can benefit from CoT.
> In appendix F we report that forcing CoT on `gemini3-pro` resulted in degraded accuracy throughout all corridor lengths, while with `gpt-4.1` it resulted in a slight improvement.

---

> > ### Comment · Reviewer_Xvrr · 2026-03-07
> > **Acknowledgement**
> >
> > I appreciate the author's revision on 1) improved precision of claims; 2) results showing that additional test-time methods still don't solve the problem. Concerns have been addressed.
> >
> > One more thought: you should see if a vision-language model can solve the problem. (Out of scope since I didn't ask for it in the original review).

---

> > > ### Author Response · Authors · 2026-03-25
> > > **Response to acknowledgement**
> > >
> > > We thank the reviewer for positive comments.
> > > As a side note we had already internally tried the idea proposed by the reviewer, by generating PNG files and feeding them with the same prompt used in the main paper.
> > > The results were inline with what reported already, at least with GPT-5-mini. We didn't try other models though.

---

### Review · Reviewer_uGeh · 2026-02-16

**Summary Of Contributions:**

The paper investigates long-horizon planning in Large Reasoning Models (LRMs) using a simplified linear Sokoban benchmark that isolates long-horizon planning from state persistence. Results show that performance deteriorates with increasing horizon length, exhibiting token growth and looping failures. Although an LLM-Modulo approach improves success rates, the study reveals persistent weaknesses in long-horizon state maintenance and planning.

**Audience:**

Yes

**Audience Explanation:**

The experiment setup provides a clean diagnostic environment for isolating long-horizon depth from other factors like branching complexity. Given ongoing discussions around reasoning scaling and planning in LRMs, such systematic assessments are likely to be of interest to the TMLR community. In particular, the paper offers empirical insights into failure modes (e.g., looping) that complement broader benchmarks.

**Claims And Evidence:**

Yes

**Claims Explanation:**

The empirical results convincingly demonstrate degradation under the specific single-pass decoding setup.

**Requested Changes:**

- The novelty is somewhat constrained by the extensive prior use of Sokoban in recent LLM reasoning benchmarking, for example in [1]. At the same time, this paper focuses exclusively on a single simplified game domain; while the controlled corridor design is clean, the broader contribution would be stronger if the methodology were validated on additional planning environments with the same purpose. Expanding experiments to another structured planning task would strengthen claims of generality and reduce the risk that conclusions are domain-specific. Comparison with other benchmarks using Sokoban to evaluate LRM reasoning should also be discussed in the paper.
- The evaluation protocol relies primarily on exact string equality with a single ground-truth sequence of actions, which conflates planning success with reproducing a specific trajectory rather than verifying goal satisfaction or optimality through a solver; this is stricter than standard planning evaluation and may incorrectly mark valid alternative or slightly suboptimal plans as failures, thereby underestimating model capability. A more appropriate approach would be to evaluate goal satisfaction independently of trajectory identity, or, the authors can explicitly verify whether the corridor setup guarantees solution uniqueness; if not, solver-based validation should replace strict string matching as the primary metric.
- The model coverage is limited and does not include more advanced reasoning models, making it unclear whether the reported limitations reflect fundamental constraints or capacity-dependent effects; without scaling analysis across model sizes, the claim that long-horizon planning deficits are structural remains premature. Including at least one state-of-the-art high-capacity model, even on a subset of tasks, would provide crucial evidence regarding whether failure thresholds shift with model size.
- The benchmark evaluates long-horizon planning using a strictly single-pass autoregressive rollout, without interactive execution feedback or step-wise replanning. Under this setup, several observed failure modes, such as repetitive looping, may be strongly influenced by long-context degradation rather than reflecting fundamental deficits in planning or search. This concern is particularly relevant for reasoning models that tend to generate lengthy intermediate traces, thereby placing additional pressure on context maintenance. As a result, it becomes difficult to disentangle architectural or context-length limitations from genuine planning competence, especially given prior evidence that model performance can deteriorate substantially when operating over very long contexts.
- The study does not control total computational budget across “reasoning-only” and LLM-modulo settings: the latter benefits from solver calls, validation feedback, and multiple retries/reformulations, while the former is evaluated in a one-shot format without reflexion/self-correction. Consequently, reported improvements under LLM-modulo are difficult to attribute purely to enhanced planning ability with the external solver (although this is clear, the experiment setup itself is weird to me).
---
[1] Shi, Jiajun, et al. "Korgym: A dynamic game platform for llm reasoning evaluation." arXiv preprint arXiv:2505.14552 (2025).

---

> ### Author Response · Authors · 2026-03-04
> **Response to reviewer uGeh**
>
> We thank the reviewer for the detailed methodological feedback, in the following we address all the excellent points raised.
>
> ---
>
> ### **1 Positioning relative to prior Sokoban work**
>
> We have expanded the related work discussion to more clearly situate the corridor benchmark within the broader Sokoban evaluation literature.
> The revised manuscript emphasizes that the purpose of the one-dimensional construction is the diagnostic isolation of horizon depth while eliminating branching complexity.
> Indeed, while Sokoban has been widely adopted as a planning benchmark for LLM reasoning, most prior work evaluates performance on heterogeneous map distributions or fixed difficulty tiers.
> In contrast, our corridor construction provides a parametrically controlled family in which planning depth scales linearly with corridor length.
> This enables systematic analysis of extrapolation limits, rotation sensitivity, and failure mode structure under controlled geometric transformations.
>
> We have updated the related works with two more references adopting Sokoban in diverse settings: "Sokoban has also been widely adopted in recent work as a benchmark for evaluating reasoning and planning in large language models, often using heterogeneous map distributions or fixed difficulty tiers (Hu et al., 2025a; Wang et al., 2025). On realistic maps most models like `Sonnet 3.7` or `OpenAI-O3` never exceed $8-10$% with most of them totally failing".
>
> Additionally, we have updated the manuscript adding a novel experiment in the Appendix H describing a different task, which we called "Freetski" as it is the free version of Klotski, the famous wooden puzzle game.
> In Freetski we have removed all the complications and branching from Klotski, by asking the LLM to simply "extract" the block from long corridors, and free it.
> Differently from Sokoban where a player, a crate and a goal exist, in Freetski only a solid block exist composed by more than one elements.
> Hence, in Freetski we require the LLM to coherently move blocks outside of long, open corridors, while respecting physical connectedness and boundaries.
>
> ---
>
> ### **2 Optimality criterion and evaluation protocol**
>
> In the revised version of the paper, we have clarified (see Section 3.4) our design choice of evaluating only the unique optimal solution. While an exponentially large number of action sequences can reach the goal in principle, such trajectories do not reflect the intended abstract reasoning in this setting.
> The core cognitive requirement of the corridor task is to infer and execute the monotonic policy `push in direction X for l steps`.
> In the revised manuscript in section we have added the following sentence
>
> > Allowing arbitrary exploratory trajectories to count as success would effectively reward environment-style trial-and-error behavior rather than structured forward reasoning, which is not the purpose of our analysis.
>
> Moreover, by means of prefix accuracy and Manhattan distance we already provide a quick metric to measure the quality of the provided solution.
>
> This distinction is particularly important because instances are information-complete and our corridor-based design have no branching factor and very importantly no deadlocks. Under these highly simplified conditions, exploratory behavior is neither necessary nor informative about abstract planning competence. For this reason, exact optimal execution remains our primary success criterion.
>
> Additionally, to directly address the reviewer’s concern, we have updated the complementary analysis based on Manhattan distance to the goal as a function of cumulative path length with `gemini3-pro` and `grok-4.1-fast` (Figure 5 of the revised manuscript).
> This diagnostic verifies whether failed trajectories respect the physical constraints of the environment.
> The results show that many failures remain physically coherent but terminate prematurely or exhibit small counting errors, reinforcing our interpretation that the dominant issue is horizon maintenance rather than misunderstanding of the dynamics.
> In the revised manuscript in section 3.4 we have updated the text as follows:
>
> > This metric, different from the accuracy with its 0/1 binary output, can provide a better idea of the adherence of the final cumulated plan result as final distance from the target, highlighting potential problems like physical constraints break or wandering behaviors.
>
> [ the response continues in the next comment]

---

> ### Author Response · Authors · 2026-03-04
> **Response to reviewer uGeh [part 2]**
>
> ### **3 Additional structured planning task**
>
> In the Appendix F of the revised manuscript we have extended our analyses on an additional structured planning task.
> The experiment evaluates the same reasoning properties in a different structured environment, which we call *Freetski*, a simplified and open variant inspired by the classical Klotski puzzle.
> In this setting, we deliberately remove the combinatorial branching and multi-object interactions characteristic of Klotski.
> We have illustrated a Freetski problem in the same appendix section.
> The task is reduced to extracting a single connected block from a long corridor and freeing it.
>
> Unlike Sokoban, which involves an agent, crates and goals, Freetski contains only one composite rigid block composed of multiple connected cells.
> The model must generate coherent movement sequences that translate this block through elongated, open corridors while preserving physical connectedness and respecting boundary constraints.
> This design isolates spatial consistency and long-horizon state tracking without the additional complexity introduced by multi-object planning or deadlock configurations.
> The results with the best performing model on Sokoban benchmark, `gemini-3-pro`, are indicative of the difficulty of structured planning tasks even in the Freetski case as they indicate that structured planning remains unstable even in the simplified Freetski setting.
> Rather than showing a smooth degradation with corridor length, accuracy becomes highly variable, especially beyond 70. Performance oscillates sharply across neighboring lengths, with abrupt drops and partial recoveries.
> Reasoning token usage increases roughly linearly with corridor length, but higher token counts do not reliably correspond to correct solutions. Extended reasoning does not stabilize performance.
> Overall, the experiment confirms that even in a minimal spatial setting, long-horizon reasoning exhibits length-dependent instability rather than robust generalization.
>
> ---
>
> ### **4 Context length versus planning degradation**
>
> The reviewer raised the possibility that observed failures may primarily reflect long-context degradation.
> The new experiments in Figure 3 help disentangle this factor. In particular, `grok-4.1-fast` exhibits sharp failure near $\\ell \\approx 40$ despite its very large context window (2M tokens claimed).
> This strongly suggests that the degradation is not explained solely by context capacity and instead reflects accumulation of execution errors over long horizons.
> Moreover, the number of reasoning tokens rarely exceeds $100,000$.
>
> We have clarified this point in the revised discussion of section 4.1.
> Apart from the addition of new models supporting our observations, we have also studied the failure modes of reasoning models systematically in section 4.2 of the revised manuscript.
> Specifically, starting from a subset of the available reasoning traces we have defined six failure modes.
> The resulting categories have then been used to instruct an LLM-as-a-judge model (`gpt-4.1`) to classify the remaining traces (more than 3000).
> We have included the prompt used to classify the traces in the appendix E.
> The failure modes we have identified are:
>
> - context overload resignation, where the model abandons precise reasoning on long or repetitive maps
> - counting errors, involving misestimation of corridor length or required moves
> - map hallucinations, where details of the layout are fabricated
> - rule misinterpretations of Sokoban mechanics
> - logical inconsistencies with impossible state transitions; and
> - formatting errors despite otherwise correct reasoning.
>
> The dominant patterns are incorrect counting and context overload resignation, rather than misunderstanding of the task rules.
>
> ### **5 LLM-modulo compute asymmetry**
>
> We have revised the description of the LLM-modulo setting to explicitly acknowledge that the computational budget is not matched to the reasoning-only configuration.
> The purpose of this experiment is diagnostic rather than comparative under equal resource constraints.
> Specifically, it isolates whether failures originate from plan generation or from execution and verification.
> The external solver serves as a controlled verification component that can validate or reject proposed plans, thereby disentangling reasoning errors from execution errors.
>
> We do not interpret the improvements observed under LLM-modulo as evidence of enhanced intrinsic planning ability.
> Instead, they reflect the benefit of external validation, iterative reformulation, and symbolic search.
> Importantly, even under this assisted regime, performance remains unstable and requires multiple retries to achieve modest gains, suggesting that underlying counting errors and long-horizon state-tracking failures persist.
>
> Section 4.3 of the revised manuscript now clearly frames LLM-modulo as an assisted upper-bound configuration rather than a budget-matched baseline comparison.

---

> > ### Comment · Reviewer_uGeh · 2026-03-20
> >
> > Thank you for the detailed response. Most of my concerns are addressed. For part 2, "Optimality criterion and evaluation protocol", I'm still a bit concerns for some cases like you need to go from upper left to bottom right (which seems to be a plausible task based on your example Figure 1), there are no unique answers about when to turn therefore better clarification of this may be useful.

---

> > > ### Author Response · Authors · 2026-03-25
> > > **Response to comment - clarifications**
> > >
> > > As specified in the paper, we utilize non-diagonal corridors with a fixed width of **1**.
> > > This design choice is fundamental to our approach because it restricts the movement to a single degree of freedom.
> > > By using the four rotational variants (0°, 90°, 180°, and 270°), we effectively constrain the agent's behavior by:
> > >
> > > - eliminating non-physical states: Since the corridor width matches the agent/object dimensions, "illegal" or non-physical movements (such as bypassing a box or clipping through walls) are architecturally impossible.
> > > - enforcing optimal path: Because the agent is confined to a 1D traversal within these corridors, any backtracking—moving away from the objective after an initial advancement—is inherently sub-optimal.
> > > - ensuring singular optimal solution: by pruning these non-physical and redundant (backtracking) movements, we ensure that the state space leads to a single, optimal sequence of actions, i.e. the precise number of steps in the required direction to reach the goal.
> > >
> > > Here are example maps to clarify the situation.
> > > The picture we have inserted in the paper was for the reader to better figure out how a Sokoban game should appear in its ASCII-based version.
> > >
> > >
> > > # 0° degrees:
> > > ```
> > > ##########
> > > #.   $  @#
> > > ##########
> > > ```
> > >
> > > # 180° degrees:
> > > ```
> > > ##########
> > > #@   $  .#
> > > ##########
> > > ```
> > >
> > > # 90° degrees:
> > > ```
> > > ###
> > > #@#
> > > # #
> > > # #
> > > #$#
> > > # #
> > > # #
> > > #.#
> > > ###
> > > ```
> > >
> > > # 270° degrees:
> > > ```
> > > ###
> > > #.#
> > > # #
> > > # #
> > > #$#
> > > # #
> > > # #
> > > #@#
> > > ###
> > > ```

---

### Review · Reviewer_soQr · 2026-02-18

**Summary Of Contributions:**

This paper proposes a minimally designed dataset to evaluate whether Large Reasoning Models can handle long horizon planning tasks. The dataset is based on a one dimensional corridor variant of the Sokoban game, characterized by a single adjustable hyperparameter ℓ and a unique optimal solution. The authors design two evaluation settings, one that does not rely on external tools and another that leverages classical PDDL planning tools. The evaluation metrics include exact action matching accuracy, prefix accuracy, and Manhattan distance. The empirical results show that the three tested models, DeepSeek R1, GPT 5 mini, and GPT oss 120B, perform well on smaller problem instances with ℓ less than about 20, but exhibit a rapid decline in accuracy as the problem scale increases. The paper also reports several interesting observations, such as token usage growing approximately linearly with problem size while incorrect samples display much higher variance in token counts. Overall, the paper presents an interesting and clean benchmark with a single controllable parameter that effectively reflects the performance of large models on long horizon planning tasks.

**Audience:**

Yes

**Audience Explanation:**

This work focuses on the long-horizon planning and reasoning abilities of LLMs, which are important for the community.

**Broader Impact Concerns:**

The broader impact of this work is largely positive, as it provides a controlled and transparent benchmark for diagnosing long horizon planning limitations in large reasoning models.

**Claims And Evidence:**

Yes

**Claims Explanation:**

The claims made in the submission are generally supported by clear and systematically presented empirical evidence. The authors construct a controlled benchmark with a single adjustable parameter, which allows them to isolate long horizon depth from branching complexity.

**Requested Changes:**

1. The paper mainly evaluates three models, DeepSeek R1, GPT 5 mini, and GPT oss 120B. While these models are representative, they may not fully reflect the most up-to-date frontier capabilities. For example, DeepSeek R1 was released about a year ago and was primarily optimized for tasks such as mathematics and code generation. Evaluating more recent reasoning models, such as newer Gemini or GPT Thinking models, would strengthen the empirical relevance of the conclusions and provide a clearer picture of current state of the art performance.

2. Puzzle tasks are an important component of reasoning model training, and many recent models may undergone reinforcement learning on similar structured reasoning problems. It would be valuable to analyze whether models that have been explicitly trained with RL on puzzle or planning tasks exhibit different behavior on this benchmark. In addition, the paper could more carefully discuss the potential impact of task-specific training or data contamination. Furthermore, the current benchmark only considers instances with a unique optimal solution. Extending the evaluation to cases with multiple optimal solutions or even unsolvable instances could provide additional insights into planning robustness and failure modes.

3. The paper reports that incorrect solutions exhibit substantially higher variance in token usage, which is a useful global statistical observation. However, it would be more informative if the authors provided a finer-grained analysis of these abnormal cases, including a taxonomy of the dominant error patterns. For example, it would be helpful to quantify how often failures are driven by inconsistency with earlier steps, repetitive looping behaviors, or other systematic breakdowns in state tracking.

---

> ### Author Response · Authors · 2026-03-04
> **Response to reviewer soQR**
>
> We thank the reviewer for the positive evaluation and helpful suggestions.
> Here we cover all the requested changes the reviewer has identified.
>
> ---
>
> ### **1 Expanded model coverage**
> Following the reviewer’s recommendation, we have integrated `gemini-3-pro` and `grok-4.1-fast` into our evaluation suite. These expanded results reinforce the empirical depth of our study and confirm that **horizon-dependent degradation** persists across diverse architectures. Figures 3, 4, and 5 in the revised manuscript now include these models alongside updated discussions. Notably, `gemini-3-pro` is now the top-performing model, demonstrating the capacity to solve corridors up to size $50$ with high accuracy. Conversely, the reasoning-heavy `grok-4.1-fast` fails at a length of approximately $40$ despite its 2M-token context window, even when configured for medium reasoning effort. As shown in Figure 10 of the revised Appendix G, `gemini-3-pro`'s performance still suffers over moderately longer horizons, falling off significantly once corridor lengths reach 100.
>
> ---
>
> ### **2 Potential training exposure**
> We investigated whether the evaluated models possess prior knowledge of Sokoban. All tested systems demonstrate awareness of the game rules and settings. However, the training data of these proprietary models are not disclosed at sufficient granularity to determine exposure to puzzle games or specific instances. We note in the paper that our corridor constructions are highly atypical and unlikely to appear in natural Sokoban corpora due to their trivial gameplay structure. While contamination cannot be fully excluded, the consistent horizon-dependent degradation across multiple frontier models suggests that memorization alone does not explain the observed behavior. We have acknowledged the impossibility to understand training exposure and mitigation strategies adopted in the “Conclusions” section.
>
> ---
>
> #### **2.1 Additional corridor variants**
>
> We agree with the reviewer that infeasible or adversarial one-dimensional layouts are a promising extension.
> We have added this direction explicitly to the future work section of the revised manuscript.
> As a note to the reviewer, we have specified in the prompt that all corridors are solvable so the model should try as hard as possible to solve them correctly, yet with varying results, as demonstrated in our work.
> We have added this and the above observations in the revised manuscript in the *Current limitations and Future works* section.
>
>
> ---
>
> ### **3 Failure mode taxonomy**
> We have added a detailed failure analysis over more than 3,000 reasoning traces in the revised manuscript (section 4.2). Errors are now systematically categorized into six distinct modes: context overload resignation, counting error, map hallucination, rule misinterpretation, logical inconsistency, and formatting error. Importantly, in the revised appendix, we also include the LLM-as-a-judge prompt used for the analysis of these failure modes.
> This disaggregated view reveals that:
> - context overload resignation and counting errors overwhelmingly dominate across all models, highlighting horizon-length sensitivity as the primary bottleneck.
> - counting errors represent the fundamental hurdle for the strongest solvers (such as `gemini3-pro` and `gpt-5-mini`), which systematically miscalculate distances despite otherwise locally coherent reasoning.
> - map hallucinations and formatting errors act as systematic, model-specific bottlenecks (e.g., prominently affecting `grok-4.1-fast` and `gpt-oss-120b`, respectively).
> - fundamental breakdowns in game logic (such as rule misinterpretations and logical inconsistencies) remain relatively rare across all conditions, indicating that explicit misunderstandings of Sokoban dynamics are not the main driver of failure.
>
> This finer-grained analysis strengthens the diagnostic value of the benchmark and directly addresses the reviewer’s request for deeper error characterization. We have added a new summary table and a picture showing the motivations of failure, with a breakdown over all the five models.

---

> > ### Comment · Reviewer_soQr · 2026-03-08
> >
> > Thank you for providing the supplementary experiments on different models, data exposure, and failure modes. I appreciate the additional analysis, and my concerns have been addressed.

---

> > > ### Author Response · Authors · 2026-03-25
> > > **Acknowledgements to the comments**
> > >
> > > We are grateful for the reviewer’s assessment. Their suggestions regarding different models and failure modes were instrumental in strengthening the paper.

---

### Decision · Action_Editor_i6yG · 2026-04-01

**Recommendation:** Accept as is

**Additional Comments:**

The paper presents reasoning limitations of large language models in a classical benchmark.

The official recommendations were generally positive. All of the official recommendations recommended acceptance, with varying degrees of strength. Comment excerpts from official recommendations:

> Please see author response to my review under "2 Optimality criterion and evaluation protocol", I feel the concern is not fully addressed and it should actually be easy to modify the evaluation to consider multiple possible paths in this grid world environment.

> This paper presents rigorous negative empirical results on simple reproducible problems, which is important to push the field further.

> This paper proposes a minimalist benchmark for evaluating the long-horizon planning and reasoning abilities of large models. The experimental results reveal the limitations and boundaries of current models. Moreover, the benchmark produces outputs that are easy to analyze: quantitative metrics can be computed directly from model outputs, and failure modes can be systematically identified to better guide future improvements.

On the point of the optimality criterion being too strict: in general, that would be true. I agree with the authors' response that for this specific paper and analysis it does not matter since the generated puzzles are all one-dimensional, and it would place emphasis something different (i.e. not structured forward reasoning). In addition, Figure 5 is a welcome addition to address this point: it uses softer metrics (distance error and accuracy) which also helps reinforce another claim made by the authors.

For the camera-ready copy, please ensure that all edits from review comments have been applied, and remove the red highlight and strikethrough markup from the current version.

**Audience:**

Yes

**Audience Explanation:**

All of the official recommendations stated that this paper will be of interest to the TMLR audience.

**Claims And Evidence:**

Yes

**Claims Explanation:**

All of the official recommendations stated that the claims are supported by the evidence.